# Tumor cell-based liquid biopsy using high-throughput microfluidic enrichment of entire leukapheresis product

Avanish Mishra[1,2,10], Shih-Bo Huang [2,3,10], Taronish Dubash[2], Risa Burr[2], Jon F. Edd[1,2], Ben S. Wittner [2], Quinn E. Cunneely[1,2], Victor R. Putaturo[1,2], Akansha Deshpande[1,2], Ezgi Antmen[1,2], Kaustav A. Gopinathan [1,2], Keisuke Otani [2,4], Yoshiyuki Miyazawa [2,4], Ji Eun Kwak[2], Sara Y. Guay[2], Justin Kelly[2,4], John Walsh[1,2], Linda T. Nieman[2], Isabella Galler[5], PuiYee Chan[5], Michael S. Lawrence [2,6,7], Ryan J. Sullivan [5], Aditya Bardia[5,8], Douglas S. Micalizzi[2,5], Lecia V. Sequist[5], Richard J. Lee[5], Joseph W. Franses[5], David T. Ting [2,5], Patricia A. R. Brunker[6], Shyamala Maheswaran[2], David T. Miyamoto [2,4,7] ✉, Daniel A. Haber [2,3,5] ✉ & Mehmet Toner [1,9] ✉

Circulating Tumor Cells (CTCs) in blood encompass DNA, RNA, and protein biomarkers, but clinical utility is limited by their rarity. To enable tumor epitope-agnostic interrogation of large blood volumes, we developed a high-throughput microfluidic device, depleting hematopoietic cells through high-flow channels and force-amplifying magnetic lenses. Here, we apply this technology to analyze patient-derived leukapheresis products, interrogating a mean blood volume of 5.83 liters from seven patients with metastatic cancer. High CTC yields (mean 10,057 CTCs per patient; range 100 to 58,125) reveal considerable intra-patient heterogeneity. CTC size varies within patients, with 67% overlapping in diameter with WBCs. Paired single-cell DNA and RNA sequencing identifies subclonal patterns of aneuploidy and distinct signaling pathways within CTCs. In prostate cancers, a subpopulation of small aneuploid cells lacking epithelial markers is enriched for neuroendocrine signatures. Pooling of CNV-confirmed CTCs enables whole exome sequencing with high mutant allele fractions. High-throughput CTC enrichment thus enables cell-based liquid biopsy for comprehensive monitoring of cancer.

Liquid biopsies provide increasingly important non-invasive strategies for longitudinal monitoring of cancer, guiding personalized therapies as tumor cells evolve under selective pressures[1]. DNA sequencing-based analyses of tumor-derived circulating DNA fragments (ctDNA) now enable the detection of drug-sensitizing and drug-resistant mutations, and they are being tested to detect minimal residual disease following surgery and early recurrence based on mutational or DNA methylation abnormalities[2–4]. However, ctDNA does not inform

[1]Center for Engineering in Medicine and Surgery, Massachusetts General Hospital and Harvard Medical School, Boston, MA 02114, USA. [2]Krantz Family Center for Cancer Research, Massachusetts General Hospital Cancer Center and Harvard Medical School, Charlestown, MA 02129, USA. [3]Howard Hughes Medical Institute, Bethesda, MD 20815, USA. [4]Department of Radiation Oncology, Massachusetts General Hospital and Harvard Medical School, Boston, MA 02114, USA. [5]Division of Hematology Oncology, Massachusetts General Hospital Cancer Center and Harvard Medical School, Boston, MA 02114, USA. [6]Department of Pathology, Massachusetts General Hospital and Harvard Medical School, Boston, MA 02114, USA. [7]Broad Institute of MIT and Harvard, Cambridge, MA 02142, USA. [8]Hematology/Oncology, University of California, Los Angeles, USA. [9]Shriners Children's Boston, Boston, MA 02114, USA. [10]These authors contributed equally: Avanish Mishra, Shih-Bo Huang. ✉e-mail: dmiyamoto@mgh.harvard.edu; DHABER@mgh.harvard.edu; mehmet_toner@hms.harvard.edu

tumor-derived transcriptional signals or cell surface protein bio-markers, whose relevance to cancer immunotherapy and antibody-drug conjugates is rapidly emerging. In addition, ctDNA measurements reflect bulk tumor-derived signals, admixed with abundant material from normal hematopoietic cells, precluding detailed analysis of tumor clonal heterogeneity at the single-cell level. Intact circulating tumor cells (CTCs) are also shed by cancers into the bloodstream, and while they provide the full complement of single-cell analytes, including high-molecular-weight DNA, intact RNA, cytoplasmic and cell-surface proteins, and metabolic markers, the rarity of CTCs in the blood has limited their clinical utility[5,6].

A standard 10 mL blood tube drawn from the peripheral circulation may yield 0 to 10 CTCs in a patient with metastatic cancer, depending on tumor histology and stage, as well as the type of assay applied to enrich CTCs[6,7]. Most studies report the absence of any detectable CTCs in 20–50% of patients with cancer[8]. Even when CTCs are readily detected within 10 mL of blood, their numbers may be too low to allow for statistically robust analytics, given the heterogeneity in the expression of cancer-relevant markers and the analytical sensitivity limits faced by most assays. Nonetheless, the emergence of highly effective antibody-based therapies directed against tumor epitopes highlights the critical unmet need for reliable cell-based liquid biopsies with quantitation of specific protein expression[9]. Needle biopsies of accessible individual metastatic lesions during the course of treatment have proven to be an important strategy to tailor therapeutic choices, but these are not readily repeated serially due to the invasive nature of the procedure[10]. Moreover, such biopsies only sample a single site of disease, which does not capture the multi-lesion heterogeneity of metastatic cancer[11]. Given the rapid turnover of CTCs and their representation of all invasive tumor deposits within a single blood specimen, a non-invasive technology that can generate sufficient numbers of intact cancer cells for reliable analysis would be highly impactful in guiding drug development and clinical treatment choices.

Leukapheresis is an established method for isolating rare cell populations from whole human blood, including hematopoietic stem cells for bone marrow transplantation or large numbers of T cells for CAR-T cell engineering[12]. In this standard clinical procedure, typically performed at a blood bank or apheresis center, blood is drawn from the antecubital vein in one arm and centrifuged through a continuous process that isolates mononuclear cells away from red blood cells (RBCs), plasma, and platelets, which are then returned to the patient through the contralateral antecubital vein[13–16]. In the context of CTC analysis, this procedure is also known as diagnostic leukapheresis[13–15,17,18] and is generally associated with less than 20 mL of net RBC loss. The sedimentation of CTCs overlaps with that of mononuclear leukocytes, making it possible to enrich these cells using standard leukapheresis parameters[15]. A typical leukapheresis procedure interrogates 3 L of blood volume per hour[19]. Due to their rapid turnover, white blood cells (WBCs) that are removed through leukapheresis are rapidly regenerated by the patient. Primary clinical criteria for tolerating leukapheresis include adequate venous access and cardiovascular stability, given the need for anticoagulation and the possibility of transient blood pressure and intravascular volume shifts[14].

Realizing the promise of isolating sufficient numbers of CTCs using diagnostic leukapheresis requires overcoming significant challenges to ultra-rare cell enrichment from large volumes of complex fluids. The diagnostic leukapheresis product (also known as a leukopak) contains a very high number of WBCs and platelets (50 to 100-fold higher than whole blood) within a large sample volume of approximately 100 mL. The FDA-approved CellSearch technology, which uses antibody-bound magnetic ferrofluids to enrich for CTCs expressing the epithelial cell surface marker EpCAM, can only process 5% of a leukopak[13,15,20]. Initial applications of this technology to leukopak samples have confirmed the expected increase in CTC capture, with a 11.5-fold increase in CTCs obtained from 5% leukopak (approximately

2% total blood volume), compared with the standard CellSearch CTC yield from a 7.5 mL blood tube (approximately 0.15% total blood volume)[13]. We previously reported a microfluidic CTC enrichment technology that does not require cell fixation and is sufficiently efficient to allow $10^4$-fold depletion of WBCs using magnetized antibodies, thereby enriching CTCs independent of individual tumor epitopes[7]. This CTC-iChip "negative depletion" technology can process 20 mL of whole blood within one hour, but like other microfluidic technologies, it is limited by clogging and reduced throughput as larger volumes and more concentrated fluids are processed[21]. To address these challenges and enable the sorting of the substantial numbers of blood cells present in an entire leukopak, we created an ultra-high throughput microfluidic platform[19]. This technology involves initial microfluidic debulking of RBCs and platelets, followed by flowing nucleated blood cells (WBCs and CTCs) through "magnetic lenses" positioned in proximity of the sorting channels, thereby enhancing magnetic forces by 35-fold and enhancing sorting throughput by 30-fold. In the initial design of this Leukapheresis-capable device, $^{LP}$CTC-iChip, we spiked individual cultured CTCs that were fluorescently tagged into control healthy donor leukopaks, achieving 86.1 ± 0.6% (mean ± SD) capture efficiencies.

In this work, we describe the first application of the $^{LP}$CTC-iChip to leukopaks from patients with diverse metastatic cancers (1–5 L of blood volume equivalent), with optimization of engineering parameters for successful clinical implementation and initial cellular and molecular characterization of unprecedentedly large numbers of CTCs from individual patients.

## Results
### Clinical cases
Seven patients with metastatic cancer undergoing treatment at the Mass General Cancer Center consented to diagnostic leukapheresis for research purposes (MGB IRB 2020P000251). Leukapheresis was performed at the Mass General Blood Transfusion Service. All patients tolerated the procedure without any reported adverse events. Leukopaks were maintained at room temperature and processed within 6 h of collection. Table 1 presents a brief clinical history of each patient.

### Parameters for diagnostic leukapheresis and leukopak characterization
The overall strategy for isolating CTCs from diagnostic leukapheresis samples collected from patients with cancer is described elsewhere[22], and it is schematically illustrated in Fig. 1. Leukopaks were generated from the seven patients with metastatic cancer, interrogating approximately one full blood volume (5.82 ± 0.96 L) (mean ± SD) over 2-h in continuous mononuclear cell collection mode (Spectra Optia) (see Methods for apheresis settings and Supplementary Table 1). Given their comparable sedimentation rate, CTCs are collected in the same layers as mononuclear leukocytes[12,13]. Diagnostic leukapheresis was performed at a flow rate of 54.5 ± 7.2 mL/min, with the volume of collected leukopaks 108.9 ± 5.3 mL (Fig. 2A, B). The WBC concentration of leukopaks was 48.7 ± 22.0 × $10^6$ cells/mL (range 21.1 × $10^6$ cells/mL to 95.8 × $10^6$ cells/mL), which is approximately 8-fold higher than that of normal whole blood. The inter-sample variability in leukopak WBC concentration was as high as 4-fold, considerably higher than the variation across the different whole blood samples. The platelet concentration of leukopaks was 1.3 ± 0.4 × $10^9$ cells/mL, approximately 4-fold higher than that of normal peripheral blood. The RBC content of leukopaks was tightly maintained, with a mean hematocrit below 2% (1.9 ± 0.2%) (Fig. 2C–H).

In addition to the high concentration of WBCs and platelets within leukopaks, their total numbers were very high, with 5.3 ± 2.3 billion WBCs and 146.3 ± 45.7 billion platelets, representing 88-fold and 49-fold, respectively, larger amounts than present within a 10 mL blood sample typically used for CTC enrichment (Fig. 2I–J).

**Table 1 | Clinical characteristics of patients**

| Patient | Diagnosis | Gender | Years Since Primary Diagnosis | Sites of Disease | Disease Status at Time of Leukapheresis | Treatment History |
|---|---|---|---|---|---|---|
| GU-1 | Metastatic Prostate Cancer | M | 19 | Bones, lymph nodes, adrenal glands, and peritoneum | Progressive disease | Localized: RT and ADT; Metastatic: ADT, darolutamide, abiraterone, cabozantinib/atezolizumab, docetaxel, cabazitaxel, lutetium-177 vipivotide tetraxetan |
| GU-2 | Metastatic Prostate Cancer | M | 10 | Bone | Progressive disease | Localized: RPLND (radical prostatectomy and lymph node dissection), ADT, docetaxel, enzalutamide; Metastatic: ADT, enzalutamide, RT, CC-94676 (AR degrader), TMEFF/CD3 bispecific antibody |
| TNBC-1 | Metastatic Triple-negative Breast Cancer | F | 5 | Lungs, lymph nodes, bone, and liver | Stable disease | Localized: Neoadjuvant paclitaxel/fluorouracil/ doxorubicin/cyclophosphamide, surgery, and RT; Metastatic: paclitaxel/carboplatin/atezolizumab, capecitabine/bevacizumab, sacituzumab govitecan, and Talazoparib |
| HCC-1 | Metastatic Hepatocellular carcinoma | M | 4 | Liver, retrocaval lymph nodes, lungs | Stable disease | Localized: gelfoam embolization; Metastatic: liver embolization/RT, atezolizumab/bevacizumab |
| HCC-2 | Metastatic Hepatocellular carcinoma | M | 3 | Lung, bone | Stable disease | Localized: RT; Metastatic: atezolizumab/bevacizumab, regorafenib/pembrolizumab |
| UM-1 | Metastatic Uveal Melanoma | M | 4 | Liver, bone, lung, soft tissue | Stable disease | Localized: RT; Metastatic: pembrolizumab, yttrium-90 irradiation of liver, tebentafusp |
| UM-2 | Metastatic Uveal Melanoma | M | 2 | Liver, bone, lung, soft tissue | Progressive disease | Localized: RT; Metastatic: tebentafusp, nivolumab/ipilimumab |

ADT androgen deprivation therapy, RT radiation therapy.

## Microfluidic CTC enrichment from patient-derived leukopaks

In testing large blood volumes spiked with cultured CTCs, we had previously applied two microfluidic devices in series: an initial debulking chip to remove RBCs and platelets (inertial separation array), followed by a magnetic lens-based high-throughput cell sorter (MAGLENS) (Fig. 3 and Supplementary Fig. 1)[22,23]. To process patient-derived leukopaks, we first applied the same microfluidic strategy designed to operate in the "negative depletion mode," i.e., removing massive numbers of normal blood cells to purify untagged and unmanipulated CTCs. The entire patient-derived leukopak specimen was incubated with biotinylated antibodies directed against the common leukocyte markers CD45 (pan-leukocyte), CD66b (granulocyte), and CD16 (monocyte) and then flowed through a microfluidic debulking chip to remove excess free antibodies, along with plasma, RBCs, and platelets (see Fig. 1). To optimally process clinical leukopaks, we replaced the on-chip stage I filter[22] with a new and separate filter chip, whose aperture size of 42 μm allows filtration of large aggregates of leukocytes or clots that form more commonly in patient-derived leukopaks (Supplementary Fig. 2).

The debulking chip itself takes advantage of inertial flow-based size separation using an array of rectangular microposts (200 μm × 50 μm × 52 μm)[22,23]. Using co-flow principles, the stream of fluid from the leukopak specimen is directed close to the wall of the microposts, where WBCs and CTCs experience a higher wall lift force due to their larger size, thereby moving further away from the wall compared with RBCs and platelets (Fig. 3A–D). The stream of fluid (3.6%) near each rectangular wall of microposts that contains RBCs and platelets is then siphoned away, a process that is repeated over 180 serial microposts across the entire array, thereby enabling highly efficient removal of RBCs (99.95%) and platelets (99.98%), while achieving a high yield of WBCs and CTCs. The numbers achieved using patient-derived leukopaks were consistent with our previous modeling studies (Fig. 3E)[22]. To achieve greater efficiency using patient-derived leukopaks, we parallelized 16 such inertial separation devices onto a plastic ^LPdebulking chip (Fig. 3A, B), and we used three to four such ^LPdebulking chips to process a single leukopak, depending on the total volume. The 16 inertial separation devices on each chip were connected together by the interconnect channels in a third layer (Supplementary Fig. 3). The channels present after the inertial separation array devices merely act as resistance channels through which the inertial separation array product flows. For patient-derived leukopaks, this optimization achieved a leukopak sample flow rate ranging from 109.5 mL/h to 146 mL/h and a total buffer flow rate from 522 mL/h to 696 mL/h.

Following leukopak debulking, we incubated the WBC and CTC suspension with 1 μm streptavidin-conjugated magnetic beads before flowing these through two parallel microfluidic MAGLENS sorters to deplete bead-bound WBCs (Fig. 3F, G). Like the control leukopaks previously studied[22], the cancer patient-derived leukopaks had a high concentration and a large total number of WBCs, requiring much higher cell processing throughput than existing state-of-the-art magnetic sorters[7,24]. The flow throughput of a continuous magnetic flow sorter is directly dependent on the magnetic field gradient in the deflection channels. Therefore, we incorporated high-permeability channels filled with soft magnetic iron particles. These channels act as "micromagnetic lenses" and amplify the magnetic field gradient 35-fold. This high magnetic gradient successfully enabled increased throughput in patient-derived leukopaks. Magnetic lenses create a field gradient as high as 15,400 T/m, compared to 440 T/m in the conventional CTC-iChip design[7,22] (Supplementary Fig. 4), with each MAGLENS sorter processing a leukopak at a flow throughput of 48 mL/h (3 billion cells/hour), 60-times higher than the conventional CTC-iChip. The magnetic lenses are lithographically positioned 70 to 80 micrometers from the sorting channels (Supplementary Fig. 5).

Given the very high number of WBCs to be depleted from patient-derived leukopaks and their heterogeneous expression of

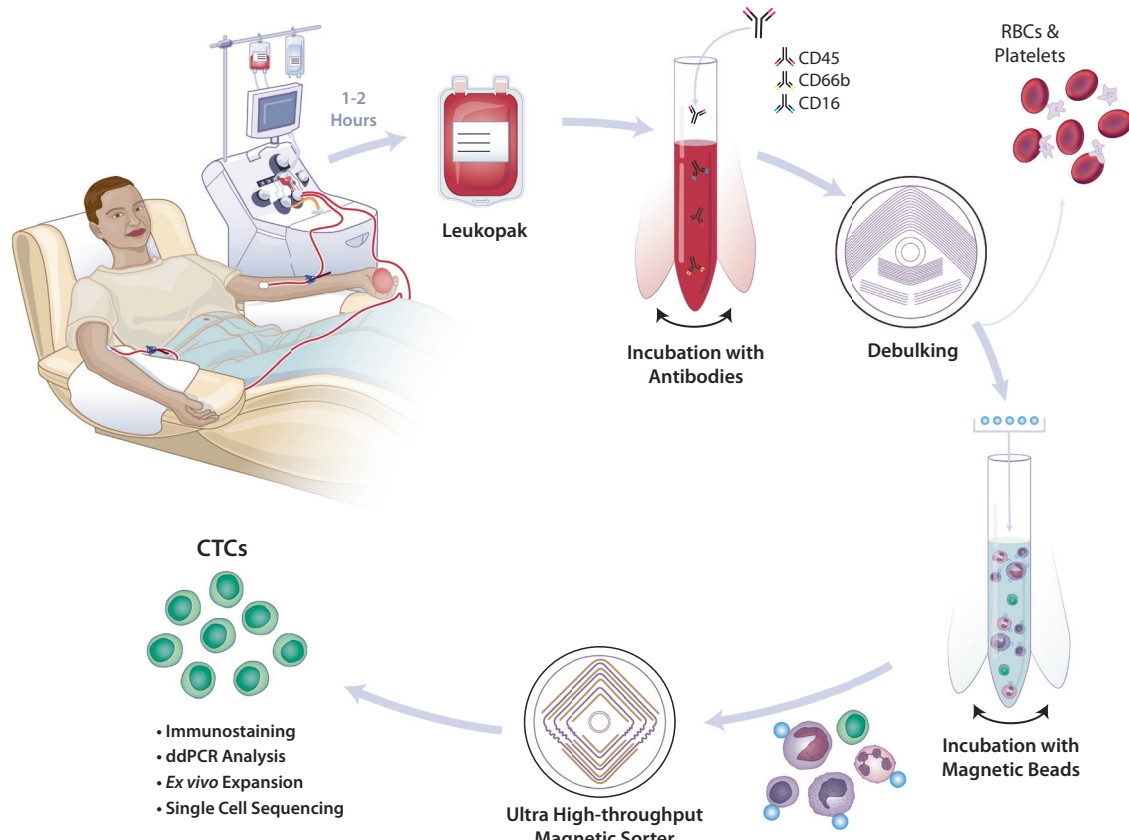

**Fig. 1 | Schematic illustration of high-volume CTC enrichment using microfluidic isolation of diagnostic leukapheresis samples.** The leukapheresis product (leukopak) is collected from a patient during a 1 to 2 h procedure that samples 3 to 6 L of blood volume. Following the addition of biotinylated antibodies to tag WBCs, a microfluidic debulking chip is used to remove unbound antibodies, RBCs, platelets, and excess plasma. Streptavidin-conjugated magnetic beads are added to tag WBCs, which are then separated from untagged CTCs using an ultrahigh-throughput microfluidic magnetic sorter. Enriched CTCs are imaged using immunofluorescent staining for lineage or tumor markers, subjected to RNA-based quantitation of specific transcripts (ddPCR), or cultured ex vivo. Single CTCs may be isolated for RNAseq or DNA analyses, including mutational profiling and CNV analyses. Together, the two chips comprise the [LP]CTC-iChip platform. Illustration by Nicole Wolf, MS, ©2024. Printed with permission.

leukocyte markers, we applied a cascaded two-stage magnetic sorting system. In stage I, cells tagged with >10 beads are deflected, and in stage II, WBCs labeled with 1 or greater magnetic beads are removed[22] (Fig. 3F–H). Figure 3I shows streak images from modeling studies where green-fluorescent WBCs are efficiently sorted from red-fluorescent CTCs. Effective depletion of WBCs carrying fewer magnetic beads requires stage II, where the flow rate is reduced through on-chip concentration to enable the depletion of cells labeled with a single bead. In stage I, the two asymmetric serpentine channels inertially focus cells in a single file by balancing shear-induced lift and Dean flow-based drag forces (Fig. 3F). The inertial focusing minimizes the possibility of WBCs colliding with a CTC and pushing it toward the center of the channel (discarded waste) during sorting (Fig. 3I). Furthermore, as leukapheresis product volumes are large, isolating CTCs within a smaller volume is essential. The inertial cell concentrator[25] allows the final product to be concentrated 11-fold. Both the co-flow in stage I and the inertial concentrators in stage II ensure that cells remain close to the side walls, where the magnetic force is strongest due to proximity to the magnetic lenses (Supplementary Fig. 4). Overall, the total flow rate across stages I and II of the MAGLENS chip, including both leukopak sample and the added buffer, is 168 mL/h.

We observed a higher microfluidic clogging risk with cancer patient-derived leukopaks than with the control healthy donor specimens previously tested[22]. The fundamental design of the MAGLENS chip greatly reduces such clogging since magnetic forces in both stages of the sorter deflect labeled cells into the center of the channel, where magnetic forces vanish (Fig. 3I). To further optimize the processing of patient-derived leukopaks, we added DNase at 100 units/mL to the cell suspension to digest any neutrophil extracellular traps (NETs), which cause fouling of the microfluidic features (Supplementary Fig. 6). Together, these strategies now allow us to achieve clog-free sorting while handling billions of nucleated cells from cancer patient-derived leukopaks.

## Multispectral imaging and cell size analysis of patient leukopak-derived CTCs

In our modeling experiments, we tested the [LP]CTC-iChip's ability to recover fluorescent-tagged cultured CTCs spiked into donor-derived leukopaks[22]. To identify unlabeled patient-derived CTCs within the microfluidic product, we applied immunofluorescence (IF)-based staining and high-content multispectral imaging, followed by digital image processing and final manual validation. CTCs were defined as being positive for both nuclear signal (DAPI) and the relevant characteristic tumor markers (see below) but negative for the cocktail of WBC markers (CD45, CD66b, and CD16). Antibodies against tumor markers used for the different tumor histologies were a cocktail of: EpCAM, pan-cytokeratin (pan-CK) and CK19 for prostate cancer (GU) and triple-negative breast cancer (TNBC)[7]; EpCAM, pan-CK, CK19, asialoglycoprotein receptor 1 (ASGR1) and Glypican 3 (GPC3) for hepatocellular carcinoma (HCC)[26]; and Sox10, melanocyte differentiation antigen (Melan-A) and neuron glial antigen-2 (NG2) for uveal

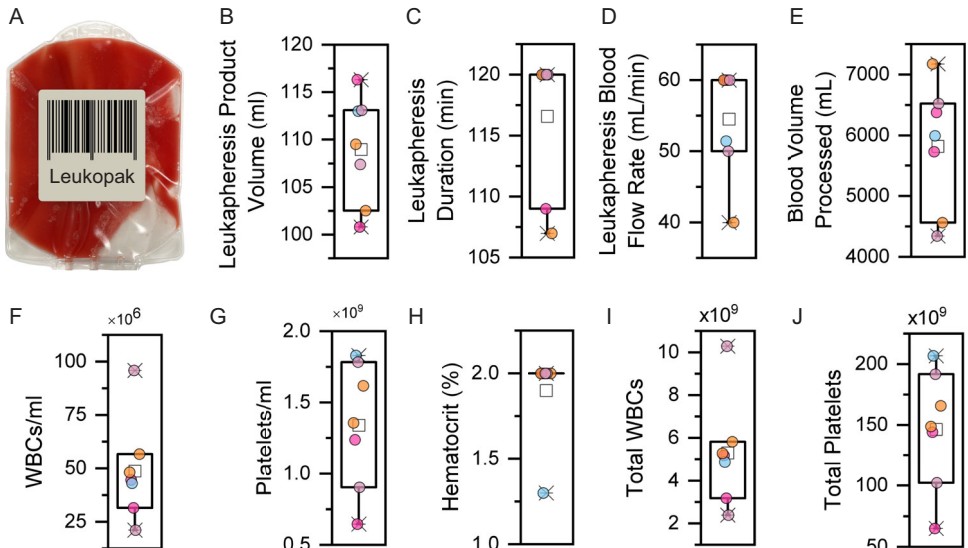

**Fig. 2 | Patient diagnostic leukapheresis parameters and cellular content in the seven cancer patient-derived leukopaks (n = 7 biological replicates).**
**A** Representative image of a 100 mL leukopak obtained from a 2-h apheresis session. **B–E** Diagnostic leukapheresis product volume (**B**), procedure duration (**C**), blood flow rate (**D**), and processed blood volume through the apheresis machine, respectively (**E**). **F–H** Concentration of WBCs (**F**), platelets (**G**) and hematocrit (RBCs; **H**) in leukopak samples. **I, J** Total number of WBCs (**I**) and platelets (**J**) in cancer patient-derived leukopaks, containing a median 88-fold more WBCs and 49-fold more platelets, compared with a standard 10 mL whole blood sample. For **B–J**, the box bounds are at 0.25 and 0.75 quartiles, the small box is centered at the mean, and the whiskers define maxima and minima. Source data are provided as a Source Data file.

melanoma (UM)[27]. Antibodies against the three WBC antigens were grouped in one fluor (Alexa Fluor 647) and those against the various tumor antigens were grouped in a second fluor (Alexa Fluor 488) (Fig. 4A–E). Using these criteria, we calculated a mean yield per leukopak of 10,057 ± 19,686 CTCs (GU-1: 58,125 CTCs, GU-2: 1276, TNBC-1: 4580 CTCs, HCC-1: 4109 CTCs, HCC-2: 1490 CTCs, UM-1: 720 CTCs, UM-2: 100 CTCs (Fig. 4F). The WBC depletion was 99.96% (3.4 ± 0.3 $Log_{10}$ depletion), thereby achieving a CTC purity ranging from 0.005% (UM-1) to 3.3% (GU-1) (Supplementary Fig. 7A, B). In GU-1, we also detected 3256 clusters of CTCs, ranging from 2–5 cancer cells tethered together in circulation (Fig. 4A) and in TNBC-1 we detected 80 two-cell CTC clusters.

To confirm the tissue of origin of enriched CTCs, we removed small aliquots, ranging from 0.5 to 1% of the final microfluidic products, to apply the previously defined digital, cell linage-based RNA signatures that capture the heterogeneity of prostate cancer (20 genes), breast cancer (16 genes), liver cancer (10 genes) and melanoma (17 genes) CTCs, while distinguishing them from surrounding blood cells with very high specificity[27–30]. In all seven cases, droplet digital PCR (ddPCR) assays confirmed the presence of the expected tumor type (Fig. 4G). As negative controls, healthy donor blood processed through the same microfluidic platform showed no ddPCR signal; diluted RNA from the relevant cancer cell lines was used as positive control.

Studies of heterogeneity across CTCs have been hampered by the small number of cancer cells isolated from individual patients using current technologies, thereby confounding inter- and intra-patient variability. The ability to isolate such large numbers of CTCs from individual patient-derived leukopaks can thus provide a vastly improved initial measure of true patient CTC diversity. Enrichment of CTCs through negative depletion of WBCs is relatively unbiased compared with positive selection of CTCs based on their expression of pre-selected epithelial markers such as EpCAM or the application of size-based selection that assumes epithelial cells to be larger than leukocytes. Indeed, morphological analysis of all scored CTCs from the patient samples, compared with their matched WBCs, shows substantial overlap in their cell diameter, with 99% of WBCs overlapping in size with 67% of CTCs (Fig. 4H). These findings are consistent with the

relatively poor yield of size-based CTC isolation across different cancer histologies. Within the many CTCs isolated from individual patients, we also note considerable variation in nuclear and cell diameters, with CTC diameters ranging from 4 μm to 25 μm (Fig. 4H). Interestingly, a bimodal distribution in both nuclear and total cell size is evident in the two HCC cases (Supplementary Fig. 7D).

We also used multispectral imaging to quantify the variability in tumor marker expression across the large number of CTCs isolated from individual patients (Fig. 4I and Supplementary Fig. 7E). The cocktail of tumor markers used to identify CTCs within each tumor type provides an initial measure of both lineage and epithelial differentiation. The intra-patient variation in tumor marker expression ranged from 3.7-fold in UM-1 to 18.8-fold in HCC-2. Similar analyses may be used to ascertain the heterogeneous expression of selected antibody-drug conjugate (ADC) targets on CTCs within individual patients in the context of personalized antibody-directed cancer therapies.

## DNA sequencing of CTCs for aneuploidy and whole exome mutational analysis

On-treatment tumor biopsies have provided invaluable information about the mechanisms whereby cancers acquire drug resistance, including new mutations, altered signaling pathways, and even cell lineage alterations. Blood-based ctDNA sequencing may reveal acquired mutations within selected cancer gene panels, but given the small quantities of low-molecular-weight DNA at low purity, it cannot achieve sufficient coverage for whole-exome sequencing, nor can it support RNA sequencing to identify altered transcriptional programs. The large number of intact CTCs isolated from leukopaks thus offers a unique opportunity to combine non-invasive blood-based diagnostics with detailed molecular interrogation of cancer cells at the single-cell level.

As a proof of principle, we selected two metastatic castration-resistant prostate cancer (mCRPC) cases, GU-1 and GU-2, with large numbers of CTCs (GU-1: 58,125 CTCs and GU-2: 1276 CTCs) making it possible to optimize molecular analyses. To readily isolate single CTCs from the microfluidic $^{LP}$CTC-iChip enriched population (GU-1: 3.3% purity, GU-2: 0.043% purity), we processed cells through a second

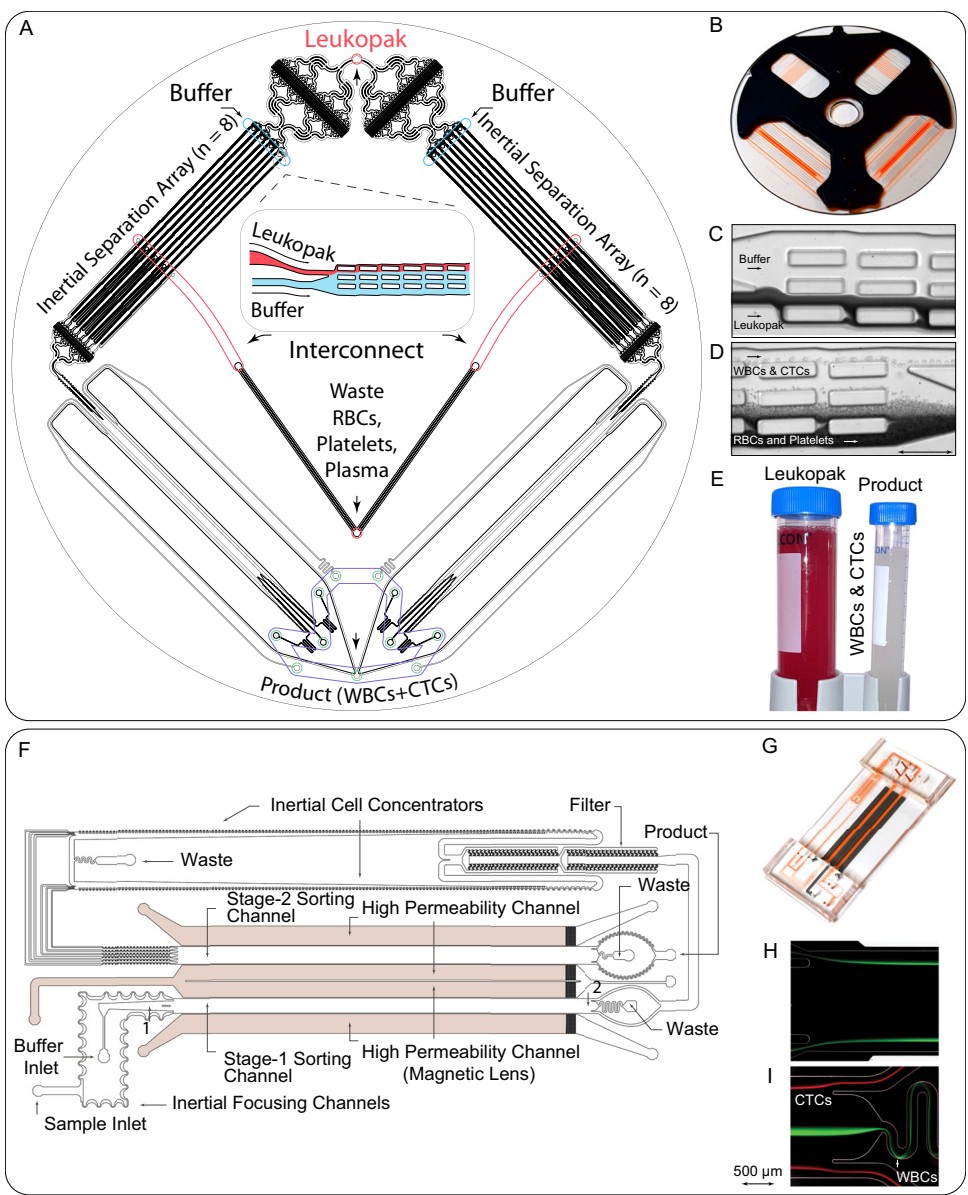

**Fig. 3 | High-throughput microfluidic devices used for processing leukopaks.**
**A**, **B** Schematic (**A**) and image (**B**) of the debulking chip used for removing unbound antibodies, RBCs, platelets, and excess plasma, concentrating WBCs and CTCs into a clean buffer. **C**, **D** Images illustrating the inertial separation of nucleated cells from RBCs and platelets within the debulking chip (*n* = 6). The underlying inertial separation array technology[23] repeatedly deflects larger nucleated cells over an array of rectangular islands (microposts) using wall lift forces to transfer the cells into a clean buffer stream, separating them from the smaller cells and non-nucleated cells (RBCs, platelets). Following the entry into the device (**C**), serial deflection across the multiple islands that comprise the Chip allows the collection of nucleated cells with very high purity and yield (**D**). **E** Image of the input sample

(leukopak) before debulking, with red color illustrating high RBC content, and of the purified product after debulking of RBCs and platelets (*n* = 6). **F** Design of the microfluidic magnetic sorter for depleting magnetically labeled WBCs, using cascaded two-stage magnetic sorting and a very high-gradient magnetic field, which is created by magnetic lenses adjacent to the cell flow channels[22]. **G** Image of the microfluidic magnetic sorter device. **H**, **I** Streak images of cells at the inlet (**H**) and exit (**I**) of stage I of the magnetic sorter (*n* = 6). Magnetic bead-labeled white blood cells (green) are deflected into the central core of the sorting channels, away from CTCs (red), thereby allowing continuous depletion of WBCs without clogging the channel. Source data are provided as a Source Data file.

purification step, applying a microfluidic fluorescence-activated cell sorter (SONY SH800) to distribute individual CTCs into single wells. We used AF488-conjugated antibodies against both the epithelial marker EpCAM and the prostate-specific markers PSMA, alongside PE-Cy7-conjugated antibodies targeting the WBC markers CD45, CD16, and CD66b, to distinguish prostate CTCs from contaminating blood cells, enabling efficient high-throughput single-cell sorting (Fig. 5A, Supplementary Fig. 8A). To further optimize the single-cell sorting process, we implemented a sequential sorting method, starting with an initial high-yield bulk sorting step of AF488-tagged CTCs to debulk residual WBCs and remove dead cells and cell fragments, followed by

sorting single cells into individual wells of PCR plates (Fig. 5B). Within the individual wells, we separated intact nuclei from single cells for DNA sequencing and the corresponding cytoplasm for RNA sequencing (RNA-seq) analysis, generating templates for paired single-cell low-pass whole-genome sequencing (LP-WGS) and single-cell RNA-seq (Smart-seq2 method)[31].

We sorted 594 individual candidate CTCs from GU-1 based on AF488 staining (EpCAM and/or PSMA) and the absence of PE-Cy7 staining (multiple WBC markers). In addition, we collected 484 "double-negative (DN)" cells, lacking staining for both AF488 and PE-Cy7. For GU-2, we sorted 192 AF488-positive cells and 192 DN cells. These

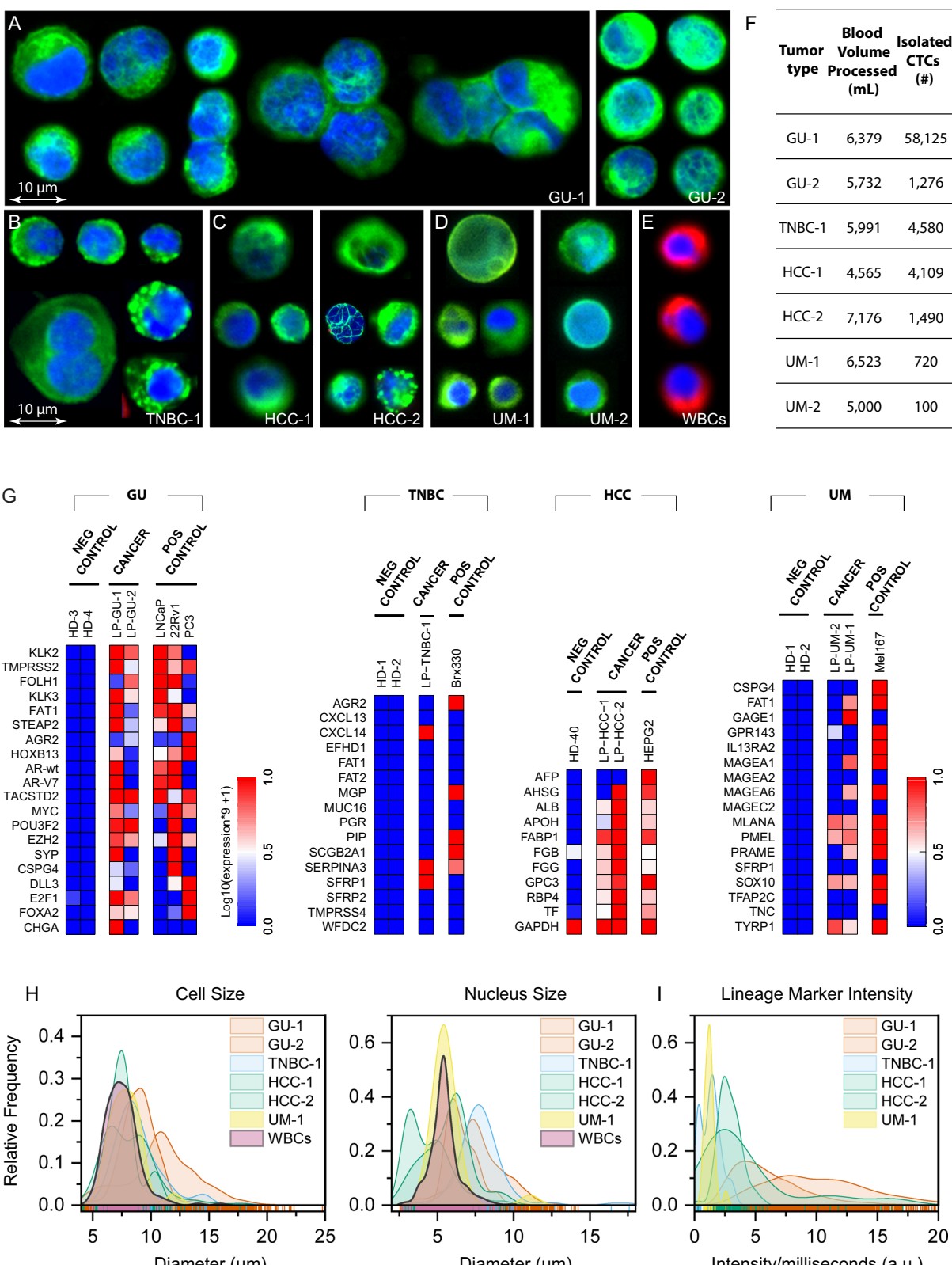

DN cells are abundant in CTC-enriched populations derived by depletion of WBC markers, but their identity is uncertain. From these single-cell collections, we selected 84 single cells from GU-1 (57 AF488-positive, 27 DN) and 173 single cells from GU-2 (115 AF488-positive, 58 DN) for LP-WGS using the MALBAC method[32] with 1.5–2× genome coverage depth, allowing for analysis of DNA copy number variation (CNV) (Ginkgo tool, http://qb.cshl.edu/ginkgo), a definitive marker of

cancer-related aneuploidy. As controls, single CTCs from ex vivo cultured breast CTCs lines[33] showed characteristic CNV, while single WBCs contaminating the microfluidic preparation showed the expected normal diploid genomes (Supplementary Fig. 8C). After filtering out samples with low-quality DNA sequencing, CNV analysis indicated high-confidence aneuploidy in 36/39 (92%) single cells expressing EpCAM and/or PSMA and in 7/10 (70%) DN cells from GU-1, and in 76/

**Fig. 4 | Analyses of microfluidic-enriched CTC bulk populations.**
**A**–**D** Immunofluorescence images of representative CTCs enriched from leukopak samples from patients with metastatic prostate cancer (GU-1 and GU-2, n = 2), triple-negative breast cancer (TNBC-1, n = 1), hepatocellular carcinoma (HCC-1 and HCC-2, n = 2), and uveal melanoma (UM-1 and UM-2, n = 2). Bulk cell populations are stained with DAPI nuclear marker (blue), the relevant tumor markers grouped within a single color (green) (see individual epitopes below), and with antibodies against the WBC markers CD45, CD16, CD66b (red). Tumor markers: **A** GU-1 and GU-2 (EpCAM, pan CK, CK19), **B** TNBC-1 (EpCAM, pan CK, CK19), **C** HCC-1 and HCC-2 (EpCAM, pan CK, CK19, ASGR1, GPC3), **D** UM-1 and UM-2 (Sox10, Melan-A, NG2). **E** Representative contaminating WBCs. **F** Table listing blood volumes processed and yield of CTCs obtained from patient-derived leukopaks (median 2799 CTCs per leukopak). **G** Droplet digital RNA-PCR (ddPCR) analysis of 0.5 to 1% of the bulk CTC products from all seven cases, quantifying expression of previously curated RNA signatures that denote either tissue lineage or cancer-specific transcripts within the background of normal blood cells. Of note, GU-1 CTCs express multiple neuroendocrine genes (CHGA, SYP, and DLL3), consistent with immunohistochemistry staining for synaptophysin and chromogranin A in a resected adrenal metastasis from this patient. Bar graphs showing expression of wild-type androgen receptor (AR-wt) and AR variant 7 (AR-V7) for GU-1 and GU-2 are shown in accompanying Supplementary Fig. S7C. Healthy donor blood is shown as negative control, with positive controls drawn from either cultured prostate cancer cell lines (LNCaP, 22Rv1, and PC3), cultured breast CTCs (BRx-142[33]), cultured liver cancer cells (HepG2), or cultured melanoma CTCs (Mel-167[66]), respectively. (H) Measured whole cell and nucleus diameters of individual CTCs (n = 5543), compared with WBCs (n = 223). Substantial overlap in size is evident between CTC and WBC populations. **I** Variation across individual CTCs from cases GU-1, GU-2, TNBC-1, HCC-1, HCC-2, and UM-1 in their intensity of staining for the combined lineage markers (see above). Source data are provided as a Source Data file.

109 (70%) single cells expressing EpCAM and/or PSMA and in 24/52 (46%) DN cells from GU-2. These alterations included shared genomic amplifications and deletions across most CTCs, as well as some unique chromosome changes consistent with tumor cell heterogeneity (Fig. 5C, D, Supplementary Fig. 9A). The CNV pattern among EpCAM- and/or PSMA-positive cells and DN cells was indistinguishable, indicating that these DN cells are genuine CTCs that have lost epithelial and prostate lineage markers. Key chromosomal alterations universally detected in CTCs from GU-1 included gains in chromosomes 1q, 7, and 8, and losses in chromosomes 1p, 5, 6, and 13, common in advanced prostate cancer, as reported in the TCGA and SU2C cohorts[34–36]. In CTCs from GU-2, we detected gains in chromosomes 5, 12, and 16p, and losses in chromosomes 4, 7q, 8p, 9, 13, and 16q, consistent with clinical karyotyping from a matched bone lesion biopsy (Supplementary Fig. 9A, B).

While the high number of CTCs recovered from prostate cancer cases enabled detailed CNV analyses, we also observed aneuploidy within single CTCs in two HCC patients (Supplementary Fig. 10). Whereas both GU-1 and GU-2 CNV analyses indicated a single dominant genomic lineage, the 12 single CTCs analyzed from HCC-1 showed two distinct subpopulations (n = 7 vs. 5), with a subset of shared core genomic amplifications and deletions, as well as clustered unique chromosome changes consistent with subclonal genomic heterogeneity (Supplementary Fig. 10C, D). The shared chromosomal alterations included gains in chromosomes 1, 6, and 8, and losses in chromosomes 4, 13, 16, and 17, prevalent in hepatocellular carcinoma[37–39].

To extend from CNV analysis to mutational analysis of prostate CTCs, we undertook deep coverage whole-exome sequencing (WES) with a sequencing depth of 100X. Since single-cell WES may not provide adequate coverage for comprehensive mutational analysis, for each patient, we pooled CTCs displaying CNV (EpCAM- and/or PSMA-positive; n = 26 for GU-1; n = 76 for GU-2) and subjected them to "pseudo-bulk" DNA sequencing. WES analysis of CTCs using stringent cutoffs (mutations in genes of the Cancer Gene Census (CGC), causing coding nonsynonymous or splice-site changes, and an allele frequency (AF) of at least 30%) revealed mutations that matched those identified using FDA-approved genetic tests, in either biopsy or ctDNA (Fig. 6A, Supplementary Data 2, 3). For example, WES analysis of CTCs from GU-1 revealed a focal homozygous deletion spanning the *PTEN* gene that had been identified in an adrenal metastasis previously excised from this patient and analyzed using the FoundationOne®CDx test but was not detectable by ctDNA analysis (Fig. 6B). WES analysis of CTCs also identified mutations at high allele fractions (Fig. 6A, Supplementary Data 3). For instance, WES of CTCs from GU-2 identified a pathogenic *TP53* missense mutation with an AF of 61.7%, compared with AF of 25% in a bone lesion biopsy and AF of 1.9% in ctDNA. Finally, across GU-1 and GU-2, WES analysis of CTCs identified high AF variants within known cancer genes, beyond those detectable using ctDNA or even

tissue needle biopsies, including premature stop codons in *ASLX2*, *IL6ST*, and *SIN3A* genes, and potential pathogenic missense mutations in genes involved in homologous recombination, cell cycle checkpoints and chromatin regulation (*ATM, BRCA1, BRCA2, CHD2, CHD4, CHEK2, FANCA*) (Fig. 6B, C).

## Single-cell RNA sequencing of CTCs revealing heterogeneous subpopulations

To address transcriptional heterogeneity within CNV-confirmed CTCs, we took advantage of paired single-cell RNA-seq and CNV analyses within individual CTCs, performing cytoplasmic RNA-seq from cells identified as CNV-positive by nuclear DNA analysis (Fig. 5A, Supplementary Fig. 8C, D). Among CTCs with definitive CNV, 30 cells from GU-1 retained sufficiently high-quality RNA for single-cell RNA-seq. In these cells with a shared, clonal CNV pattern, unsupervised clustering analysis revealed two distinct hierarchical clusters based on RNA expression (Fig. 7A). Gene set enrichment analysis (GSEA) identified a striking upregulation of Fibroblast Growth Factor Receptor (FGFR) signaling in Cluster-1, while Cluster-2 showed upregulation of pathways associated with inflammatory responses, chemokine signaling and IL/JAK/STAT signaling (FDR ≤ 0.25) (Fig. 7B, Supplementary Data 4). Additionally, Cluster-1 demonstrated activation of androgen receptor (AR) signaling (Fig. 7C). Similarly, two distinct hierarchical clusters were identified from 74 CTCs in patient GU-2 sharing a clonal CNV pattern (Fig. 7D). Cluster-1 exhibited upregulation of pathways associated with disease progression in castration-resistant prostate cancer (CRPC), including AR, MYC, and oxidative phosphorylation, while Cluster-2 showed striking enrichment in gene sets related to ion channels and neuroendocrine differentiation (Fig. 7E, F, Supplementary Data 4). Of note, a metastatic tumor biopsy in patient GU-2 did not capture the neuroendocrine differentiation. Thus, through the combination of sampling all sites of disease and high-resolution molecular analyses, single-CTC RNA-seq may reveal intra- and inter-patient heterogeneity in disease progression and therapeutic resistance mechanisms[40–51].

## Enrichment of small epithelial-negative prostate CTCs with neuroendocrine features

As noted above, single-cell CNV analysis in both GU-1 and GU-2 identified DN cells, lacking both epithelial and hematopoietic markers yet sharing the clonal CNV pattern of the more abundant EpCAM- and/or PSMA-positive CTCs. These cells were particularly abundant in GU-2, with RNA-seq identifying them within the neuroendocrine-like Cluster-2 (Fig. 7D, E, F). In GU-2, compared with EpCAM- and/or PSMA-positive CTCs, the DN CTCs had greatly elevated gene signatures associated with ion channels, neuroendocrine differentiation, and epithelial-to-mesenchymal transition (EMT)[52] and reduced expression of AR, MYC, and oxidative phosphorylation signaling pathways (Fig. 8A–D, Supplementary Data 4, 5). Remarkably, these DN CTCs were dramatically

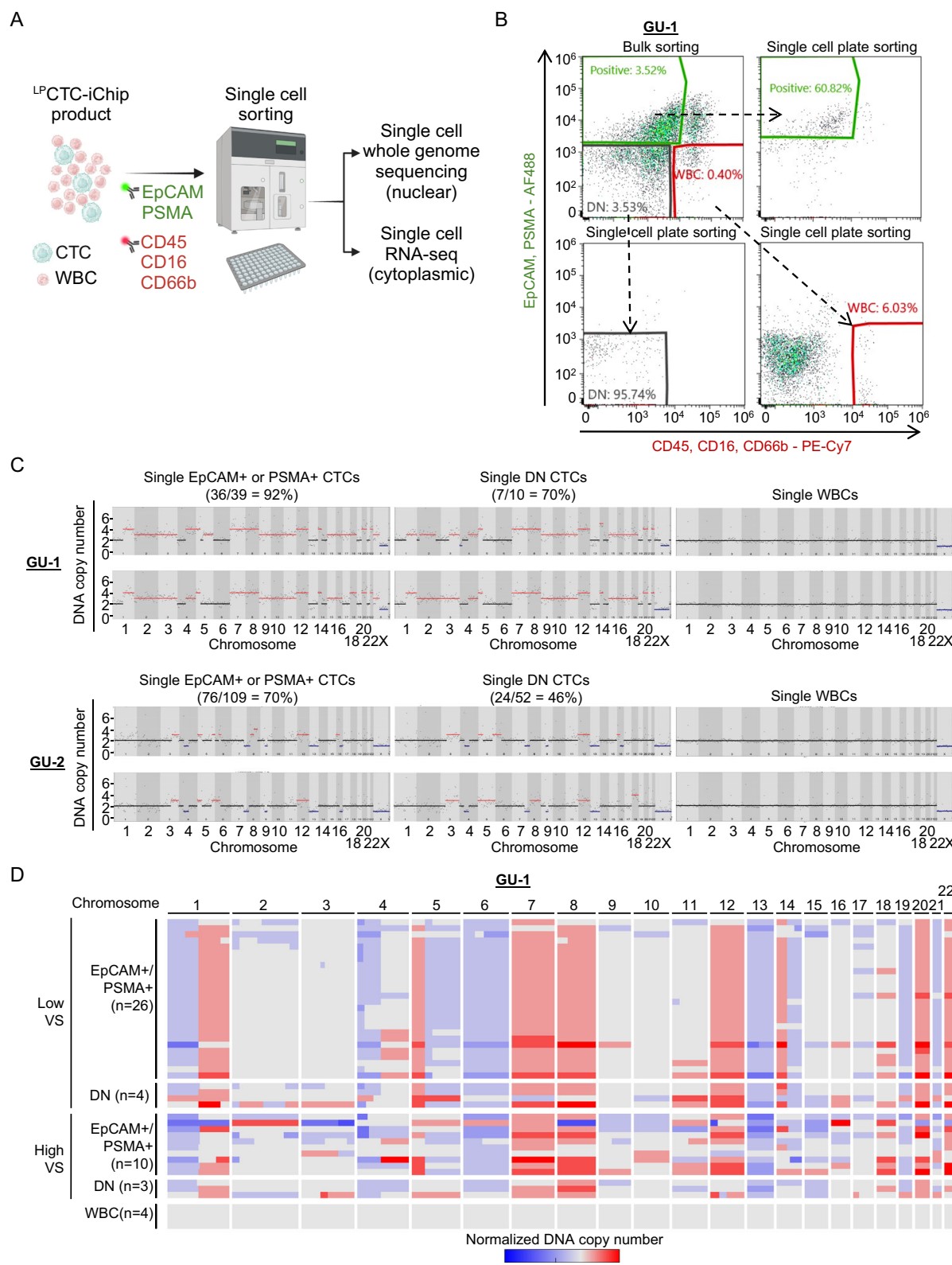

smaller in size compared with EpCAM- and/or PSMA-positive CTCs, completely overlapping with the size of WBCs (Fig. 8E, F). Neuroendocrine transdifferentiation in advanced prostate cancer has been documented with significant clinical and therapeutic implications, but its frequency based on biopsies of single metastatic lesions has been controversial[53,54]. Our observations confirm that a subset of CTCs transform into small, mesenchymal-like cells with neuroendocrine

features, apparently coexisting at a moment in time with a larger population of classical lineage-denoted prostate CTCs. Such small, transformed CTCs would escape detection using either EpCAM-targeting or size-based CTC enrichment platforms. Their identification as a specific subset of CTCs in patients with metastatic prostate cancer highlights the potential insights into tumor progression and drug resistance that may be derived from high throughput,

**Fig. 5 | Single-cell whole-genome sequencing and DNA copy number analyses.**
**A** Schematic of single-cell isolation from [LP]CTC-iChip-enriched leukopak samples using Fluorescence Activated Cell Sorting (FACS). The CTC-enriched [LP]CTC-iChip product is free of magnetic-conjugated antibodies against WBC markers CD45, CD16 and CD66b. For FACS single-cell isolation, two-color separation is achieved using an AF488-conjugated antibody cocktail against EpCAM and PSMA (for patients GU-1 and GU-2; green) and a PE-Cy7-conjugated antibody cocktail against CD45, CD16, CD66b (labeling contaminating low-expressing WBCs that escaped [LP]CTC-iChip depletion; red). Individual cells (CTCs and WBCs) are then subjected to paired single-cell whole-genome sequencing and single-cell RNA-seq. Figure was created using BioRender (Agreement number: DF26ZQNZ7S). **B** Two-step FACS-sorting strategy using an initial ultra-yield bulk sorting to remove dead cells and debris, followed by single-cell plate sorting. A representative single-cell sorting profile is shown here, isolating candidate CTCs that are positive cells for either EpCAM (epithelial marker) or PSMA (prostate lineage marker); cells of uncertain identity that are negative for both EpCAM and PSMA, as well as the WBC markers CD45, CD16 and CD66b (Double-negative; DN); and white blood cells that are positive for CD45, CD16 and CD66b. **C** Representative DNA copy-number variation (CNV) analysis in individual CTCs, compared with diploid WBCs from the same patient (X chromosome haploid in male patient). Cancer cell-associated aneuploidy was confirmed by CNV in 36/39 single CTC candidates (EpCAM- and/or PSMA-positive) and in 7/10 single DN cells from patient GU-1 and in 76/109 single EpCAM- and/or PSMA-positive CTCs and in 24/52 DN cells from patient GU-2. Ginkgo was used for DNA copy-number analysis from single-cell whole-genome sequencing data. **D** Normalized heatmap shows clustering of single-cell CNVs across the genome in CTCs ($n = 43$) from patient GU-1. All CTCs show shared core chromosomal alterations, while WBCs are diploid. Variability score (VS) quantified DNA and assay quality (low VS corresponds to high quality). Source data are provided as a Source Data file.

tumor epitope-agnostic CTC enrichment, and detailed molecular characterization.

## Discussion

We have described the clinical application of a high-throughput microfluidic device for efficient and semi-automated enrichment of large numbers of CTCs from leukopaks drawn from patients with metastatic cancer. CTC analyses have had limited clinical utility, given the extreme rarity of cancer cells in the blood circulation of patients with cancer and the fact that current technologies can only process 10 to 20 mL of blood for CTC enrichment. As such, the ability to consistently obtain large numbers of cancer cells through microfluidic enrichment of an entire leukopak presents a unique opportunity to transform both research and clinical applications of CTCs in cancer.

### Microfluidic processing of large cell numbers and volumes

Leukapheresis is routinely used in clinical settings to obtain sufficient numbers of hematopoietic stem cells for bone marrow transplantation or T cells for CAR-T engineering[12,19]. In these applications, large quantities of normal hematopoietic mononuclear cells are enriched through centrifugation via leukapheresis, followed by immunological selection, achieving purities of >90% and without requiring downstream single-cell analytics. In contrast to hematopoietic cells, CTCs are extraordinarily rare in the bloodstream, and they are often pre-apoptotic from loss of cell adhesion signals, shear stresses of blood flow, and high oxygen tension within the vasculature, making them susceptible to further destruction ex vivo. The robust isolation of large numbers of CTCs from patient-derived leukopaks thus presents a major technological challenge, both in terms of ultra-rare cell detection as well as the need for low-shear-stress microfluidics[55]. Furthermore, the most important insights to be derived from CTC analyses arise from single-cell-level molecular analyses, in which tumor heterogeneity at the DNA, RNA, and protein levels can be assessed.

Increasing the amount of blood volume interrogated has long been perceived as key to CTC-based diagnostics[8,56]. To date, a number of studies have applied diagnostic leukapheresis in patients with either metastatic or localized cancer, demonstrating the feasibility of the clinical procedure and the presence of abundant CTCs within leukopaks[13–16,57–59]. Most studies used the FDA-approved CellSearch CTC enrichment platform, based on batch processing magnetic capture of EpCAM-positive cells, to enrich CTCs from leukopaks. While increased CTC yields were reported in breast, pancreatic, lung, and prostate cancers, the CellSearch assay can only process 5–10% of the entire leukopak[13,15,16], limiting the total number of isolated CTCs. Other approaches for CTC enrichment, including RosetteSep-based leukocyte depletion[14,18], density gradient centrifugation[60], RBC lysis[17] followed by bulk magnetic separation, and size-based separation[57], have also been deployed for processing larger volumes of diagnostic leukapheresis samples. Patterns of CNV in single or pooled CTCs have

been demonstrated[13,20], and RNA profiles have been reported following 10X or Nanostring-based sequencing[17,60]. Compared with batch-purification technologies, continuous-flow processes through microfluidic platforms have the considerable advantage of ultra-rapid and very gentle processing, making it possible to rapidly sort through an entire leukopak while preserving CTC integrity. Microfluidic processing also does not require cell fixation, thereby enabling the high-quality single-cell paired RNA and DNA analyses described here.

From a technological standpoint, microfluidic processing of leukopaks for ultra-rare cell detection is hampered by the large total number of WBCs (approximately 6 billion cells), their high concentration (8-fold higher than normal blood), and the large total fluid volumes (100 mL). Clogging of channels, resulting from platelet aggregation or release of DNA NETs from lysing WBCs, is a major obstacle (Supplementary Fig. S6), as is target cell loss from the massive numbers of non-target cells to be removed and associated prolonged processing time. Two essential features of the [LP]CTC-iChip make efficient sorting of large cell volumes possible. First, both components of the chip are designed to deflect blood cells away from the walls, thereby minimizing the possibility of clogging despite the very high concentrations of WBCs and platelets. In the debulking chip, the cell-free region near the walls of rectangular pillars is created by wall lift forces (Fig. 3C, D). In the MAGLENS sorter, the magnetic field directs magnetically labeled cells into the core of flow at the center of the channel, away from the side walls (Fig. 3I). This is made possible by the coplanarity of the opposing magnetic lenses on either side of the sorting channels, with a symmetric force towards the center of the channel, such that the magnetic gradient gradually vanishes as cells near the center of the channel (See Supplementary Fig. 4). As a consequence, the sorted cells are moved into the core of the Poiseuille flow, creating an inherently clog-free, safe design that can process billions of cells and a large fluid volume.

Second, the MAGLENS sorter employs a two-stage sequential strategy to deplete WBCs, a critical consideration since WBCs are highly variable in their expression of the cell surface markers used to bind magnetic-conjugated beads. As such, WBCs with high expression of CD45, CD16, or CD66b acquire large loads of magnetic beads on their cell surface during labeling, and these cells may clog channels upon application of a strong magnetic force. The chip design thus involves an initial deflection of WBCs with a large magnetic load through a high-flow channel in Stage I, followed by the passage of remaining WBCs with lower epitope expression and magnetic load through a low-flow channel in Stage II. Approximately 90% of WBCs are removed in stage I, with the rest in stage II, achieving a 99.96% total depletion without compromising the flow rate or increasing the risk of clogging.

We note that the [LP]CTC-iChip platform enriches CTCs through "negative depletion" of WBCs and is, therefore, tumor epitope-agnostic. This is illustrated in the tumor types selected for analysis

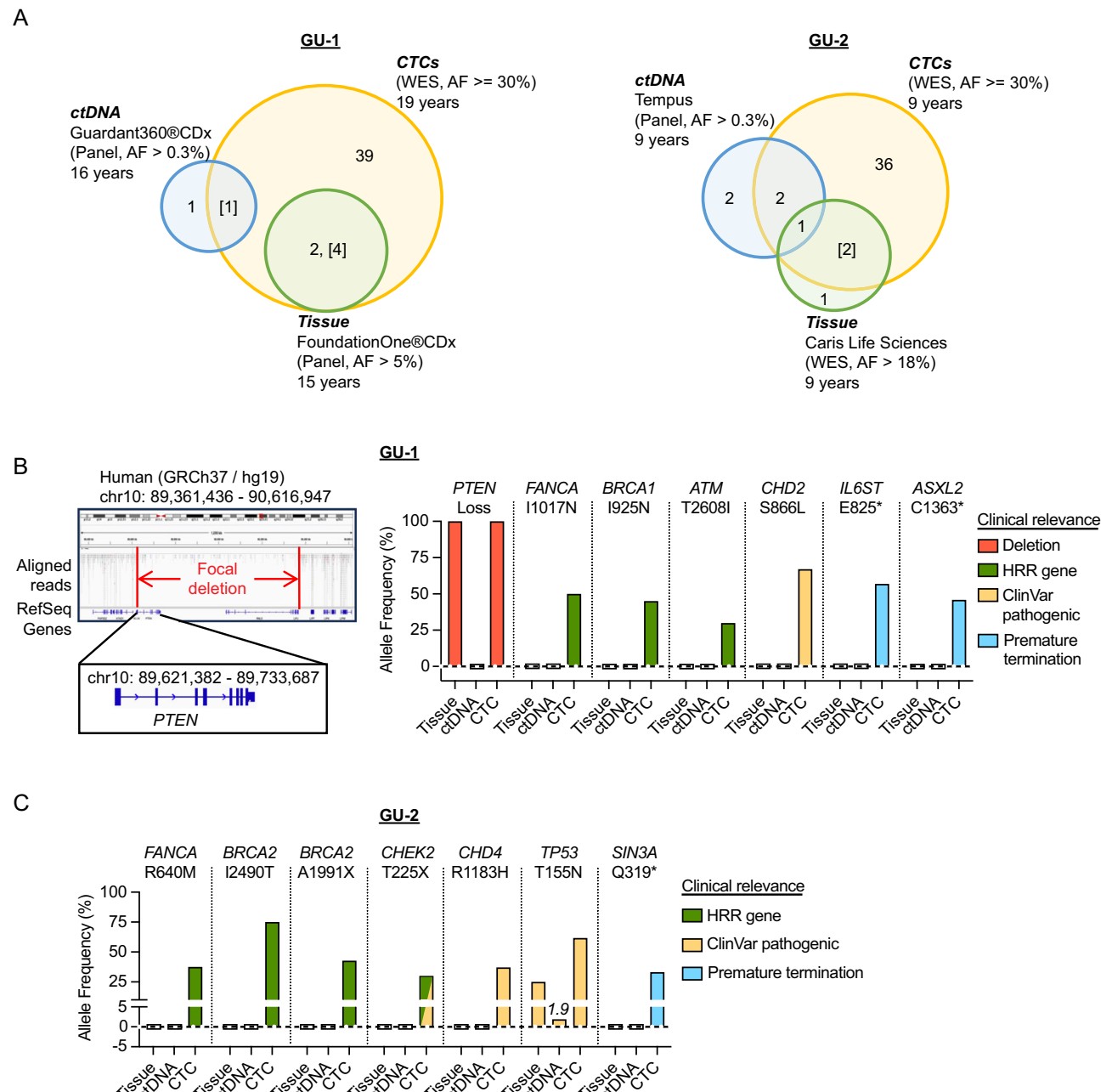

**Fig. 6 | Whole-exome sequencing (WES) data analysis for prostate CTCs displaying CNVs. A** Venn diagrams show the number of mutations reported in the matched metastatic lesions and ctDNA, as identified by WES analysis of CTCs in patients GU-1 and GU-2. Assay types (cancer gene-specific panel or WES) with name of FDA-approved test are shown, along with allele frequency (AF) cut-off for variant calling, and the number of years since initial diagnosis at which the different tests were obtained. WES analysis of CTCs was analyzed using MuTect and Strelka software with stringent cutoffs, including 30% allele frequency (see Methods for details). Variants of unknown significance (square brackets) found in matched biopsy tissue or ctDNA were also identified in WES analysis of CTCs by relaxing cutoff settings to an AF of a least 1% with at least two supporting reads and including mutations in genes outside the Cancer Gene Census. **B** The left panel shows IGV visualization of a deletion spanning the PTEN gene, identified by WES analysis of CTCs. The PTEN deletion was also detected in a matched surgical resection of a metastatic tumor lesion in patient GU-1. The right panel (Boxplot) shows mutations with potential clinical impact, identified by cancer panel genotyping of metastatic lesion biopsy or ctDNA or by WES analysis of CTCs from patient GU-1. Relative allele frequencies are shown for mutations known to be pathogenic, as reported in ClinVar, premature termination mutations, or mutations in homologous recombination repair (HRR) genes. A hollow rectangle hovering around 0 indicates an allele frequency of 0%, meaning no variant was detected by genetic tests. **C** Boxplot shows similar mutations identified in metastatic lesion biopsy, ctDNA, or WES analysis of CTCs from patient GU-2. The TP53 mutation T155N was scored by all three analyses, albeit with variable allele fractions: CTC (61.7%), tumor biopsy (25%), and ctDNA (1.9%), as displayed above the bar. A hollow rectangle hovering around 0 indicates an allele frequency of 0%, meaning no variant was detected by genetic tests. Source data are provided as a Source Data file.

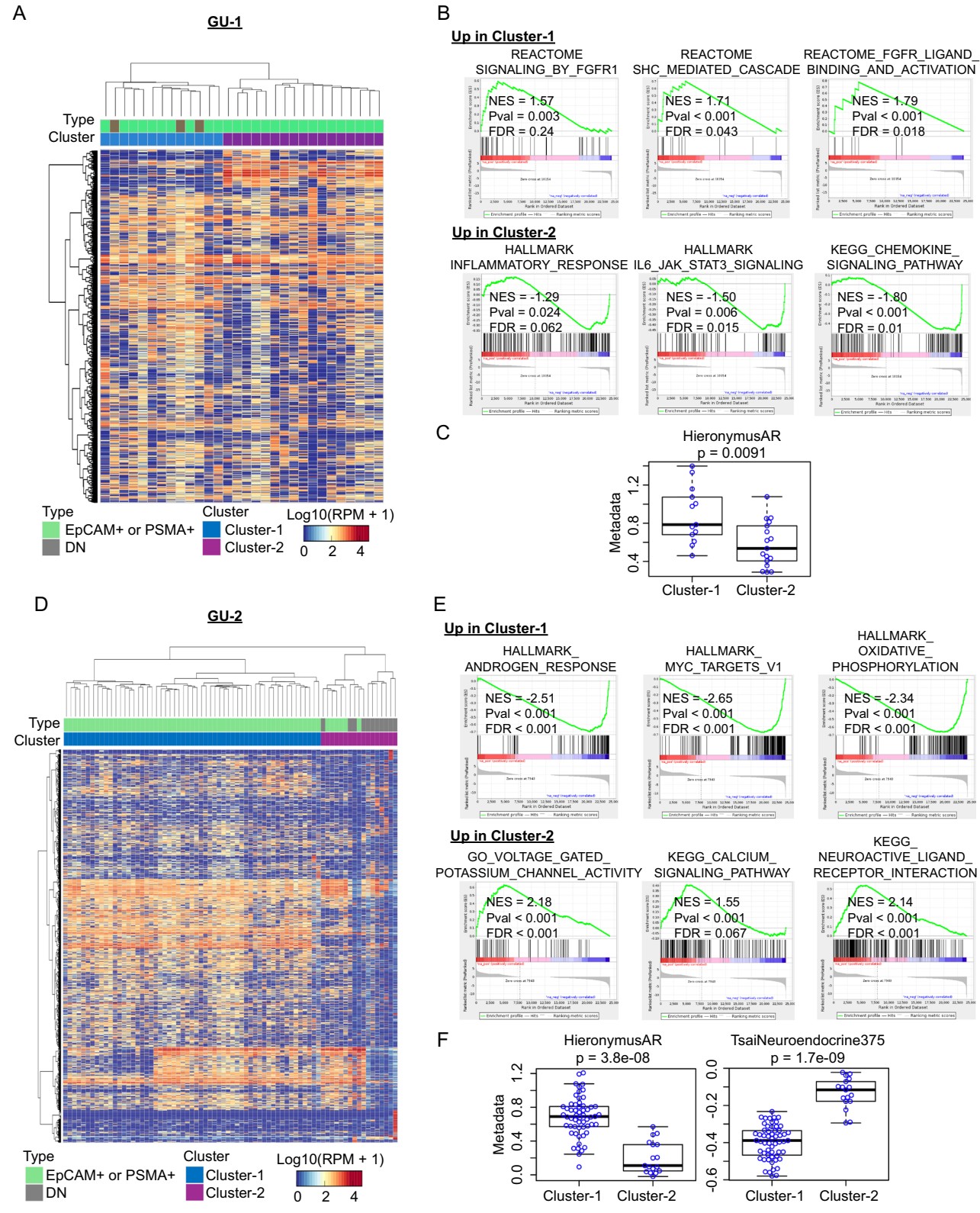

here: melanomas, including uveal melanomas (UM), have minimal expression of traditional epithelial markers such as EpCAM, and both hepatocellular cancers (HCC) and triple-negative breast cancers (TNBC) also have relatively low EpCAM expression. For these tumor types, the initial epitope-agnostic enrichment is followed by CTC scoring by staining with a cocktail of antibodies targeting multiple shared and lineage-specific epitopes. Nonetheless, the ᴸᴾCTC-iChip

platform used here is also adaptable to a "positive selection" mode, including the capture of CTCs by virtue of their expression of EpCAM or other epithelial or tumor-specific epitopes, which may be valuable in specific tumor-targeting contexts.

While discrimination between rare CTCs and abundant surrounding WBCs using antibody-mediated detection has limitations, including variability in cell surface protein expression, it appears well

**Fig. 7 | Single-cell RNA-seq data analysis for prostate CTCs displaying CNVs. A** A heatmap showing unsupervised hierarchical clustering of single-cell RNA-seq data reveals two distinct expression clusters of CTCs from patient GU-1. The three DN CTCs fall within Cluster-1. **B** Gene set enrichment analysis (GSEA) shows elevated FGFR signaling in Cluster-1 of CTCs from patient GU-1 ($n = 13$; three different signatures), whereas Cluster-2 ($n = 17$) shows enrichment in inflammatory signaling pathways. The permutation-based p-values, normalized enrichment score (NES), and FDR values are from GSEA software. **C** Comparison of metagene expression shows upregulation of androgen-responsive genes (Hieronymus AR[50] in Cluster-1 of CTCs from patient GU-1. **D** A heatmap showing unsupervised hierarchical clustering of single-cell RNA-seq data reveals two distinct expression clusters of CTCs from patient GU-2. The 11 DN CTCs fall within Cluster-2. **E** GSEA shows upregulation of signaling pathways associated with disease progression in castration-resistant

prostate cancer (CRPC) in Cluster-1 of CTCs ($n = 57$) from patient GU-2, whereas Cluster-2 ($n = 17$) shows elevated ion channels and neuron-related pathways. **F** Comparison of metagene expression shows upregulation of AR signaling (Hieronymus AR[50]) in Cluster-1, whereas Cluster-2 (containing DN CTCs) shows elevated expression of genes associated with neuroendocrine differentiation (Tsai Neuroendocrine $n = 375$ genes[42]). For **B** and **E**, correction for multiple hypothesis testing is given by FDR values. GSEA calculates p-values and FDR values based on permutation testing. If GSEA does not observe any false discoveries for a gene set in the default 1000 permutations, p-values and FDR values as <0.001 were reported. For **C** and **F**, the thick horizontal bar is at the median, thin horizontal bars are at 0.25 and 0.75 quartiles, and whiskers are at 1.5 x interquartile range. p-values were computed using a two-sample, two-sided Wilcox test with no adjustment for multiple hypothesis testing. Source data are provided as a Source Data file.

suited for large-volume enrichment strategies. Approaches to CTC isolation that rely on the reported large size of CTCs compared with WBCs are confounded by the significant overlap between these cell populations, as illustrated here using an enrichment platform that is unbiased by cell size. Established ATCC tumor cell lines are typically larger than WBCs, but primary CTCs in the bloodstream appear to be heterogeneous in size, even within individual patients, with 67% of CTCs overlapping in size with 99% of WBCs (Fig. 4H). In addition, important subsets of CTCs, including the double negative neuroendocrine-lineage prostate CTCs described here fall entirely within the WBC size range. Other non-antibody-based approaches to CTC analysis, such as direct plating of blood cells onto a surface followed by high-speed imaging[61], cannot be readily scaled up by 100-fold to process the number of cells within a leukopak. Taken together, microfluidic platforms have the advantage of being readily scaled up and automated[7], with the potential to produce a point-of-care device for clinical applications in cancer monitoring.

## Clinical applications of CTC enrichment from diagnostic leukapheresis samples

Liquid biopsies, primarily centered on DNA mutation analysis from ctDNA, have emerged as powerful tools for monitoring cancer evolution during treatment, including the emergence of mutations that confer resistance to targeted therapeutic drugs[1]. However, the advent of antibody and epitope-specific therapies brings with it a pressing need to measure cell-surface protein markers on tumor cells. Most recently, the successful deployment of antibody-drug conjugates, which target cell surface epitopes on cancer cells to deliver chemotherapy payloads, is dependent on the expression of the appropriate cell surface protein, as are bispecific antibodies that recruit T cells to cancer cell epitopes[9,62]. Stratifying patients for treatment with the correct therapeutic antibody will be critical, given the rapidly increasing armamentarium targeting different cell-surface epitopes, as well as on-treatment monitoring for the emergence of resistant cancer cells, some of which may have lost expression of the targeted epitope[63]. Combined with the large number of CTCs harvested by negative depletion of WBCs from leukopaks, the multispectral CTC imaging platform will yield a highly quantitative single-cell assessment of epitope expression without biased selection for EpCAM-positive cancer cells. As with all liquid biopsies, CTC measurements also have the advantage of sampling multiple sites of disease rather than a single biopsied lesion, and they can be repeated serially and non-invasively.

In addition to the quantitation of specific cell-surface epitopes for immune therapies, the ability to interrogate large numbers of intact cancer cells from the blood in individual patients offers unprecedented opportunities for serial multi-omics at the single-cell level, including paired RNA and DNA analyses applied to large numbers of individual CTCs. In the prototype cases of prostate cancer described here, leukopak-derived CTC analysis using WES reproduced the mutations identified by cancer gene mutation panels in either biopsied

metastatic lesions or blood-based ctDNA sampling (e.g., *PTEN* and *TP53*), and also discovered additional potentially significant mutations in genes implicated in homologous recombination (HRR), cell cycle checkpoints and chromatin remodeling. These findings are highly clinically relevant as PARP inhibitors are indicated for the treatment of patients with HRR-deficient prostate tumors. Moreover, the very high allelic fraction derived from CTC analyses may enable distinction between heterozygous and homozygous mutations, which is important in assessing such mutations and which is not achievable with the very low allelic fractions in most ctDNA sequencing analyses.

Paired single-cell DNA and RNA sequencing makes it possible to perform detailed transcriptional analyses on CNV-confirmed CTCs. In the prostate cancer cases described here, matched DNA and RNA analyses uncovered previously unsuspected and distinct cancer cell subpopulations, sharing clonal CNV patterns, but exhibiting divergent signaling pathways. In one case (GU-1), one subset of CTCs was driven by FGFR signaling and the other one with inflammation-associated signatures. In the second case (GU-2), an AR-driven CTC subset coexists with a second cancer cell population exhibiting neuroendocrine differentiation. Such RNA-defined heterogeneity was not identified in either clinical case through metastatic tumor biopsies or ctDNA analyses, and their detection through CTC sequencing illustrates the potential for early detection of emerging drug resistance mechanisms. Of note, paired single-cell DNA and RNA sequencing also made possible the detailed molecular analysis of "double negative" DN CTCs, which comprise CNV-confirmed prostate tumor cells having lost epithelial markers and transformed into small, mesenchymal-like cells with neuroendocrine features. While this degree of lineage transformation in prostate cancer is thought to be infrequent based on tumor biopsies, real-time CTC analyses in patients with advanced prostate cancer may be needed to determine the frequency with which classical and neuroendocrine prostate cancer cells coexist within individual patients.

While diagnostic leukapheresis, coupled with high-throughput CTC enrichment, presents a strategy for clinical deployment of a whole-cell-based liquid biopsy, apheresis requires specialized clinical facilities, which imposes a practical hurdle, compared with the collection of a standard 10 mL blood draw. The patients with metastatic cancer who underwent diagnostic leukapheresis for this study did not have procedure-related complications, nor have such been reported in other diagnostic leukapheresis studies[13,16,17,20]. Nonetheless, it is likely that minimally invasive leukapheresis-based CTC analyses will be ideally suited for specific clinical decision points, and the fractional blood volume to be sampled may be reduced in cancer types with high CTC burdens.

Finally, we note the importance of single-cell CNV analysis in the evaluation of CTCs. Although even rarer than CTCs, some non-malignant epithelial cells and reactive stromal cells may be shed along with cancer cells from the tumor microenvironment, and rare endothelial fibroblasts or other non-hematological and non-cancerous cells may be harvested, particularly in high-volume blood screening[64,65].

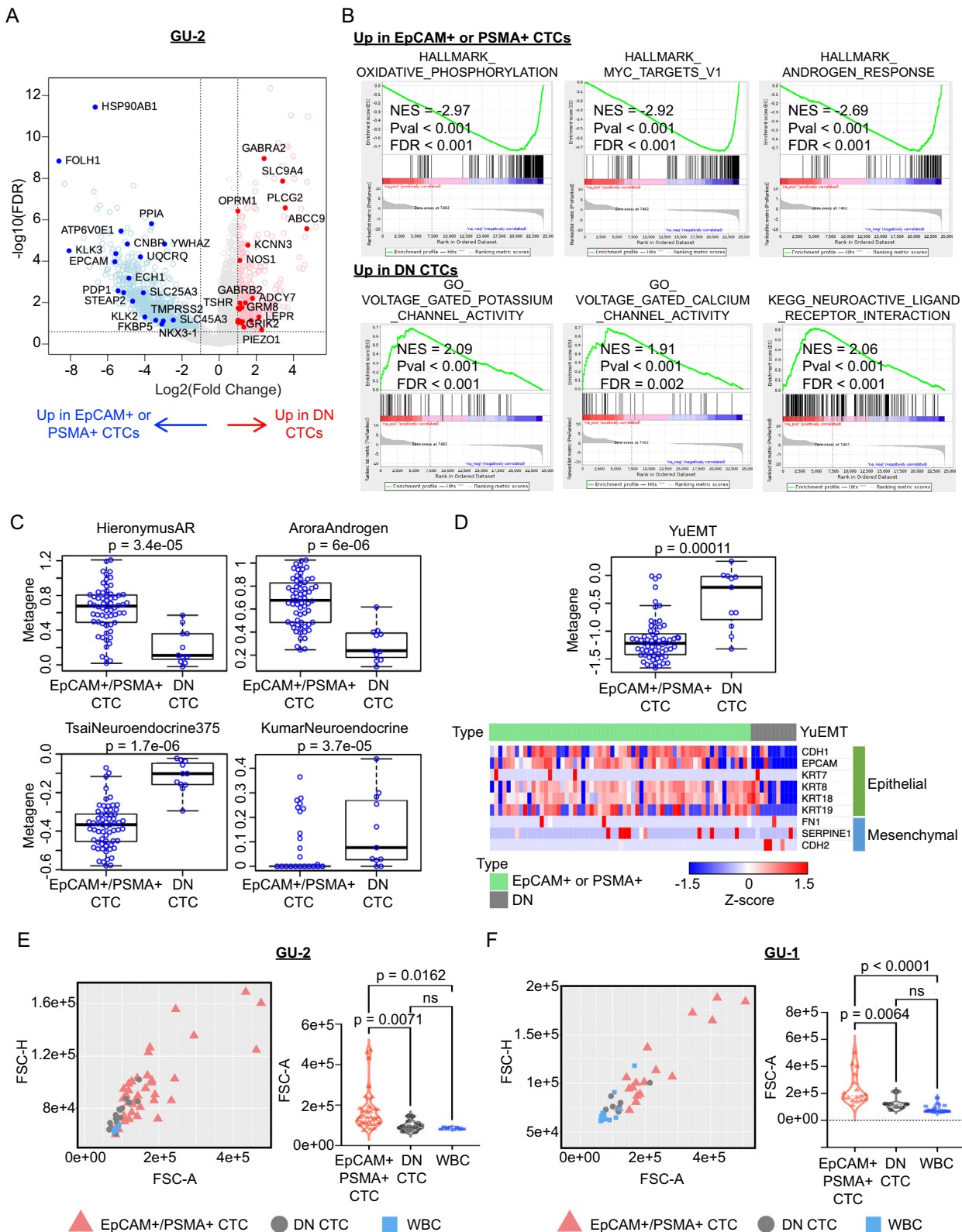

Single-cell CNV analysis provides the most efficient and compelling identification of a tumor cell since aneuploidy is a unique and defining characteristic of cancer, and it is readily measured using low-pass NGS sequencing without requiring prior knowledge or interpretation of point mutations that may be of uncertain significance. In this context, the combination of high-throughput CTC screening and definitive CNV measurement may have important applications in the blood-based

detection of localized cancer. Molecular signatures of CTCs are currently detectable within 10–20 mL of whole blood using microfluidic enrichment in approximately 10–20% of patients with localized prostate or breast cancers[30,34]. While not examined here, leukopak-based screening in patients with localized cancers have also reported dramatic increases in CTC yield[15,16]. Robust CNV determination may prove to be a definitive companion test for emerging ctDNA-based cancer

**Fig. 8 | Identification of small "double negative" (DN) prostate CTCs with neuroendocrine markers. A** Volcano plot highlighting differentially upregulated genes in DN CTCs ($n = 607$) versus canonical EpCAM- and/or PSMA-positive CTCs ($n = 2006$) from patient GU-2. Representative upregulated genes associated with AR, MYC, oxidative phosphorylation pathway in EpCAM- and/or PSMA-positive CTCs are denoted by blue filled-in circles with gene names, and representative upregulated genes associated with ion channels and neuron-related pathways in DN CTCs are denoted by red filled-in circles. **B** GSEA reveals upregulation of signaling pathways associated with disease progression in CRPC in EpCAM- and/or PSMA-positive CTCs, whereas DN CTCs exhibit elevated ion channels and neuron-related pathways. GSEA calculates permutation-based p-values, correction for multiple hypothesis testing FDR values and normalized enrichment score (NES). p-values and FDR values were reported as <0.001, as GSEA did not observe any false discoveries for a gene set in the default 1000 permutations. **C, D** Comparison of metagene expression shows (**C**) high AR signaling (Hieronymus AR[50]; Arora Androgen[45]) in EpCAM- and/or PSMA-positive CTCs (n = 63) and high neuroendocrine feature (Tsai Neuroendocrine $n = 375$ genes[42]; Kumar Neuroendocrine[43]) in DN CTCs ($n = 11$) from patient GU-2, and (**D**) high epithelial-to-mesenchymal transition (Yu EMT) in DN CTCs from patient GU-2. The z-transformed heatmap also shows the expression of epithelial markers in EpCAM- and/or PSMA-positive CTCs, while DN CTCs lose expression of epithelial markers and gain mesenchymal markers (Yu EMT[52]). The thick horizontal bar is at the median, thin horizontal bars are at the 0.25 and 0.75 quartiles, and whiskers are at 1.5 × interquartile range. p-values were computed using a two-sample, two-sided Wilcox test with no adjustment for multiple hypothesis testing. **E, F** Scatter plots and violin plots show the small cell size of DN CTCs isolated from patients GU-1 and GU-2, which is comparable with that of WBCs. The DN CTCs are considerably smaller in size than patient-matched classical EpCAM- and/or PSMA-positive CTCs. Statistical significance was determined by one-way ANOVA. Adjustment for multiple hypothesis testing was done by the post-hoc Tukey test. p-values higher than 0.05 are considered not significant "ns". Source data are provided as a Source Data file.

screening methods based on the interpretation of altered DNA methylation or fragment sizes. Looking to the future, the development of CTC-based RNA analytics will provide exceptional opportunities to understand heterogeneous subpopulations of cancer cells, monitor their evolution following interventions, and even assess transcriptional outputs of on-target drug effects[28].

In conclusion, through the clinical application of a high-volume and high-throughput microfluidic platform, the $^{LP}$CTC-iChip, we have demonstrated the ability to purify large numbers of CTCs from diagnostic leukapheresis products in patients with metastatic cancer. The technology is feasible, scalable, and can be developed as a point-of-care technology, ultimately enabling a genuine "cell-based liquid biopsy" with multi-analyte analyses applicable to a diverse array of cancers. Leukapheresis-based CTC analysis may enable many clinical applications that have previously been limited by the small numbers of CTCs present in 10-20 mL of blood and allow a comprehensive approach to liquid biopsies that will help pave the way for more effective and personalized cancer management strategies.

## Methods

Our research complies with all relevant ethical regulations. Patient samples were collected under Mass General Brigham (MGB) Institutional Review Board protocol # 2020P000251 and research was conducted under the Mass General Brigham Institutional Biosafety Committee protocol # 2020B000186.

### Microfabrication

The Magnetic Sorter chips were designed and produced at Massachusetts General Hospital (MGH) using soft lithography with Polydimethylsiloxane (PDMS). Briefly, this involved applying a layer of SU-8 50 at 2200 rpm for 30 seconds onto a silicon wafer that had been pre-baked. Photolithography was then utilized to pattern channels by exposing the SU-8 layer to 365 nm UV radiation through a mylar mask. The exposed silicon wafer was subsequently processed with Baker BTS-220 SU-8 developer to form microfeatures.

To create a 1-mm thick PDMS layer, a Sylgard 184 PDMS kit was mixed in a 7:1 ratio of base to cross-linker and poured onto the SU-8 mold. After degassing within a vacuum chamber, the PDMS layer was cured in a convection oven at 65 °C for 12 h. Given the device's thickness of only 1 mm, inlet and outlet ports for press-fit tubing connections were established using 3 mm thick rectangular PDMS pads. These thicker PDMS pads were generated by depositing PDMS onto a featureless SU-8 50 mold in a 10:1 ratio of base to cross-linker. These rectangular pads were attached to the device via plasma bonding, and holes were punctured using a 1.2 mm biopsy punch. Subsequently, the device was bonded to a 3-inch × 1.5-inch glass slide cleaned with piranha solution, followed by baking at 85 °C for 10 min. This was succeeded by an extended bake for 3 h at 150 °C to augment the bonding and Young's modulus of the PDMS.

A manifold was designed to house the magnetic sorter chip, accommodating six permanent magnets arranged in two parallel quadrupolar configurations. This manifold was machined from 6061 aluminum alloy, with screws securing the magnets in place[22].

Debulking chips were fabricated with medical-grade cyclic olefin copolymer (COC) using variotherm injection molding by Stratec Biomedical AG (Germany). Each debulking device included 16 inertial separation array devices.

### Apheresis collection

Diagnostic leukapheresis on cancer patients was performed at the MGH Apheresis Center using an experimental protocol reviewed and authorized by the MGB Institutional Review Board (MGB IRB 2020P000251). Leukopaks were collected on approximately one human blood volume using the Spectra Optia system in the continuous mononuclear cell collection mode. All seven patients tolerated apheresis well, and no adverse event was reported. On average, the apheresis took 2 h to complete (Supplementary Table S1). The gender of the study participants was collected when provided as self-reporting data, and it is outlined in Table 1. Informed consent was obtained from study participants. Sex and/or gender was not considered in the study design. We have reported corresponding disaggregated numbers for individual patients in the main manuscript and the Source Data file. Whole blood donations from healthy donors at our lab were collected using MGB IRB Protocol # 2009-P-000295. In some cases, whole blood samples from healthy donors were also procured from Research Blood Components, LLC (Brighton, MA).

### WBC depletion antibody preparation

The WBC depletion cocktail included three antibodies: biotinylated anti-human CD45 (Thermo Fisher Scientific, clone HI30, IgG1, 0.25 μg/million cells), biotinylated anti-human CD16 (BD Biosciences, clone 3G8, IgG1, 0.05 μg/million cells), and biotinylated anti-human CD66b (Novus Biologicals, clone 80H3, IgG1, 0.025 μg/million cells). This antibody cocktail was prepared by mixing antibodies in sterile filtered 1X phosphate-buffered saline (PBS) and 0.1% BSA and stored at 4 °C for use within 7 days.

### Magnetic beads preparation

To label WBCs, we employed 1 μm Dynabeads MyOne Streptavidin T1 superparamagnetic beads from Invitrogen. The beads underwent three rounds of magnetic washing using 0.01% Tween 20 in 1X PBS, followed by three additional washes with 0.1% BSA in 1X PBS to minimize nonspecific binding. Subsequently, these streptavidin-conjugated beads were stored at 4 °C in a 0.1% BSA solution at 10 mg/mL. Washed beads were utilized within a week.

## Cell lines

Breast cancer CTC lines (BRx-142, BRx-68, BRx-330[33]) and melanoma CTC line (Mel-167[66]) were internally established at Massachusetts General Hospital by our research group. Cancer cell lines LNCaP, 22Rv1, HepG2, and PC3 were sourced from ATCC.

## Single-cell sorting

The enriched leukopak by [LP]CTC-iChip were Fc blocked (Jackson ImmunoResearch Laboratories, Cat# 009-000-008) and then immunostained with AF488-conjugated EpCAM (Cell signaling, Cat# 5198) / PSMA (Invitrogen, Cat# MA5-18161) / GPC3 (Novus Biologicals, Cat# NBP2-47763AF488) / FITC-conjugated ASGPR1 (Novus Biologicals, Cat# NBP1-51109) antibodies and PE-Cy7-conjugated CD45 (Biolegend, Cat# 982310) / CD16 (Biolegend, Cat# 980110) / CD66b (Biolegend, Cat# 396910) antibodies, and LIVE/DEAD Red (Invitrogen, Cat# L34971). Cells were treated with DNase I (Worthington Biochemical Corporation, Cat# LS006361) to reduce cell clumping in single-cell suspensions before cell sorting. We performed two-step sorting methods (the ultra-yield bulk sorting followed by the single cell plate sorting) to isolate single viable CTCs (EpCAM- and/or PSMA-positive for prostate cancer (Fig. 5B, Supplementary Fig. S8A) and EpCAM-, GPC3-, and/or ASGPR1-positive for liver cancer (Supplementary Figs. S8B, S10)) and WBCs (CD45/CD16/CD66b-positive) using SONY sorter SH800. The single viable cells were individually sorted into 96-well PCR plates containing the cell lysis buffer for downstream single-cell whole genome sequencing and RNA sequencing.

## Paired single-cell WGS and RNA-seq

For these experiments, we used either single CTCs or WBCs individually sorted by SONY sorter from [LP]CTC-iChip enriched leukopak. These were subjected to single-cell whole genome sequencing and RNA-seq analysis to obtain DNA copy number and gene expression at the single-cell resolution. Briefly, single cells were first lysed in 5 μl of single-cell lysis buffer, and 0.5 μl of Magnetic MyOne Carboxylic Acid Beads (Invitrogen, Cat# 65011) were then added to each single cell lysate to facilitate segregation of nucleus versus cytoplasm[67]. After magnetic separation, the pellet (aggregated beads with the intact nucleus) was resuspended in 5 μl of DNA lysis buffer and subjected to single-cell whole genome amplification using the MALBAC protocol[32], while the supernatant containing cytoplasmic RNA was converted to cDNA and amplified using Smart-seq2 protocol[31]. The sequencing libraries were constructed using Nextera XT DNA library preparation kit (Illumina, Cat# FC-131-1096) and sequenced on the Illumina NovaSeq 6000 platform with 150 bp paired-end reads and 1.5–2× genome coverage.

## WGS pipeline

Reads were trimmed using TrimGalore (https://www.bioinformatics.babraham.ac.uk/projects/trim_galore/) version 0.4.3 with default settings. Reads were then aligned to version hg19 of the human genome using version 0.7.15 of bwa-mem[68] with default settings. We applied the MarkDuplicates function of the Picard library (http://broadinstitute.github.io/picard) and then discarded from the resulting BAM file any reads for which the UNMAP, SECONDARY, QCFAIL, DUP, or SUPPLEMENTARY flags were set. We then used the bamToBed program of version 2.18.2 of bedtools and uploaded the resulting bed files to the Ginkgo website (http://qb.cshl.edu/ginkgo). We processed the samples using Ginkgo's default settings, except "mapped with" was set to "bwa," and for single cells, "bin size of" was set to 5 Mb.

## Copy number heatmap

We used the copy number values in the SegCopy spreadsheets that can be downloaded from the Ginkgo website. To compensate for Ginkgo's tendency to sometimes assign copy numbers that are too high, for each sample we subtracted the median copy number from the copy number. Then we added 2 to form a Normalized DNA copy number and capped the copy numbers to 5. To cluster the capped Normalized DNA copy number profiles, we used hierarchical clustering as implemented by the hclust function of version 3.6.3 of R with the Ward.D agglomeration method and Euclidean distance.

## Variability score

We computed a variability score (VS) as described in the supporting information for Knouse et al.[69] with the exception that we used 1-Mb bins instead of 500-kb bins. We considered samples with this VS greater than 0.85 to be of low quality.

## RNA-seq pipeline

Raw fastq reads generated from the HiSeq X sequencer were first cleaned using TrimGalore (v0.4.3) to remove the adapters from reads and remove reads with low sequencing quality. Cleaned reads were aligned to the human genome (hg19) using Tophat (v2.1.1)[70]. PCR duplicates were removed using samtools (v1.3.1)[71], and gene counts were computed using HTseq (v0.6.1)[72]. To cluster the RNA-seq profiles, we used the 2000 genes with the highest standard deviation across the samples. We then used hierarchical clustering as implemented by the hclust function of version 3.6.3 of R with the ward.D agglomeration method and distance defined by Pearson's correlation coefficient.

## Differentially expressed genes

To identify the subset of transcripts differentially expressed in CTCs, we first discarded genes with no variation. Then we performed a two-tailed variance-equal t-test on the RPM for each of the remaining genes. The p-values from those t-tests were used to generate a false-discovery rate (FDR) estimate for each gene by the Benjamini-Hochberg method. For each gene the log2(fold-change) was computed as the average(log2(RPM)) for DN samples minus the average(log2(RPM)) for high in EpCAM- and/or PSMA-positive CTCs. Differentially expressed genes were selected based on a fold change of 2 or greater and an FDR of less than 0.25.

## Gene Set Enrichment Analysis (GSEA)

We first removed genes with 0.9 quantile of reads-per-million (RPM) less than 10. We then ran version 2.0.14 of the UCSD/BROAD gsea2 software[73] in preRanked mode on the BIOCARTA, GO, HALLMARK, KEGG, MIR, PID, REACTOME, and TFT gene sets from version 7.1 of the MSigDB (https://www.gsea-msigdb.org/gsea/msigdb/). The preRanked values were sign(logFoldChange) times -log10 of the var.-equal t-test p-value for the two classes being compared.

## Metagene value calculation

For Metagene analyses, we used the following algorithm: let $u_i$ be the log10(RPM + 1) value for the $i_{th}$ UP gene in the signature and let $d_i$ be the log10(RPM + 1) value of the $i_{th}$ DOWN gene in the signature. We defined the metagene value as $(u_1 + u_2 + u_3 + ...+ u_n - d_1 - d_2 - ... - d_m)/(n + m)$ were n is the number of UP genes in the signature and m is the number of DOWN genes in the signature.

## WES and mutation analysis

We pooled MALBAC-amplified genomic DNA of 26 EpCAM- and/or PSMA-positive CTCs from patient GU-1 and 76 EpCAM- and/or PSMA-positive CTCs from patient GU-2 displaying clear CNV as the pseudo-bulk DNA. Exomes were captured using Agilent SureSelect Human All Exon V6 Kit and sequenced on the Illumina NovaSeq 6000 platform with 150 bp paired-end reads and 100x coverage. We ran the pipeline described above for WGS. We then ran MuTect (v1.1.7) and Strelka (v2.9.10) on the resulting BAM to detect mutations (union of MuTect and Strelka) and small indels (Strelka), and employed the following four filters: (1) the mutation must not be found in a panel of normals (PoN) based on The Cancer Genome Atlas (TCGA)[74], (2) the mutation must be in a gene that is in the Cancer Gene Census (https://www.

sanger.ac.uk/data/cancer-gene-census), (3) the mutation must be protein-altering, i.e. coding nonsynonymous or splice-site (according to GENCODE gene definitions), and (4) the mutation must have an allele frequency of at least 30% and be supported by at least 3 reads.

### Cytospin and immunostaining

For immunofluorescence-based CTC enumeration, a part of the [LP]CTC-iChip product was fixed using 0.5% PFA for 10 min, followed by cytospin at 2000 rpm for 5 min using the Thermo Scientific Cytospin 4 centrifuge and Epredia EZ Megafunnel. After cytospin, slides were washed using cold 1X-PBS, followed by nonspecific blocking for 1 h with 3% BSA and 2% Normal Goat Serum solution. This was followed by staining for 1 h with primary antibodies for nucleus markers, tumor markers, and leukocyte markers, followed by washing with 1X-PBS. Please refer to Supplementary Table S2 for a complete list of antibodies used in this work. Slides were subsequently coverslipped using a mounting medium (ProLong Diamond Antifade Mountant with DAPI from Life Technologies) and left to cure for 12 to 16 h before imaging.

### CTC enumeration method

Prepared slides were scanned using an Akoya Vectra Polaris™ Automated Quantitative Pathology Imaging System. High-content multispectral images were taken at 40X magnification in DAPI, AF488, and AF647 channels, with exposure times in each channel determined from prior spiked cells, healthy donors, and whole blood patient samples. These immunofluorescence images were captured using an 8-bit camera, which divides pixel intensity from 0 to 255. The images were subsequently exported and processed using a HALO image analysis platform. First, automated analysis of the entire slide was used to identify individual cells and the levels of fluorescent signal in each channel. Adhering to preset inclusion and exclusion parameters, digital image processing-based segmentation algorithms marked a subset of these cells for further manual verification. These cells were then individually analyzed, considering overall morphology, size, and staining intensity. Following independent analysis by at least two researchers, CTCs were scored and quantified.

### Statistics & reproducibility

No statistical method was used to predetermine sample size. No data were excluded from the analyses. The experiments were not randomized, and the investigators were not blinded to allocation during experiments and outcome assessment.

### Droplet Digital PCR analysis

Droplet Digital PCR (ddPCR) to detect the expression of published cancer-specific gene marker panels[27–30] was performed on leukopak-derived RNA. For the hepatocellular carcinoma and breast patient samples, whole transcriptome amplification (WTA) was performed on RNA extracted from 0.5% of the leukopak product using the SMARTer Ultra Low-input RNA kit, version 4 (Takara). During WTA, 18 cycles of amplification were performed. For the uveal melanoma patient samples and prostate cancer patient samples, reverse transcription was performed on RNA extracted from 0.5% of the leukopak product using the SuperScript III First-Strand Synthesis System (Life Technologies). Gene-specific nested PCR amplification was employed for 10 or 14 PCR cycles, depending on the gene, for the uveal melanoma patient samples before ddPCR. No amplification was performed for prostate cancer patient samples. Samples were prepared for ddPCR following the standard protocol (BioRad). Briefly, 2× ddPCR Supermix for probes (no dUTP) was combined with 1–2% of the patient cDNA and 1× primer/probe mix (IDT). Please refer to Supplementary Data 1 for a complete list of probes used in this work. After droplet generation, samples were thermocycled for 45 cycles with an annealing temperature of 70 °C and an extension temperature of 53 °C. As a negative control, healthy donor blood was also analyzed on the leukopak. As a positive control for assay function, 0.1% of cancer cell line RNA was spiked into healthy donor RNA prior to PCR amplification and downstream processing.

### ddPCR Data Analysis

The number of cDNA copies per reaction, as calculated by the BioRad QuantaSoft or QX Manager software, was used to generate heatmaps. Briefly, the signal was normalized within each marker such that the maximum value was 1 (red). For the UM and TNBC patients, a threshold was drawn at two standard deviations higher than the median healthy donor background signal for a marker, and this value was subtracted from the marker value for all samples. The subtracted value for each marker is adjusted to 0 when the result is negative.

### Reporting summary

Further information on research design is available in the Nature Portfolio Reporting Summary linked to this article.

## Data availability

Source data are provided with this paper. RNA-seq raw fastq files and read counts are available at Gene Expression Omnibus (GEO) accession number GSE255889. WGS and WES raw fastq files generated in this study have been deposited in the European Genome-Phenome Archive (EGA) with accession number EGAS50000000723 and EGAS50000000724, respectively. These data are available under restricted access to protect patient information due to IRB policy; access can be obtained upon request to corresponding authors. The processed WGS and WES data are in the Supplementary Data and Source Data files. Source data are provided with this paper.

## Code availability

No custom code or mathematical algorithm was developed for this study.

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

## Acknowledgements

We are grateful to the patients who donated blood to enable this work. We also thank the clinical and research staff at the Mass General Blood Transfusion Service and Cancer Early Detection and Diagnostics Program. This work was supported by the National Institute of Biomedical Imaging and Bioengineering (P41EB002503), the National Cancer Institute (U01CA214297, U01CA268933, R21CA260989, and R01CA255602), the National Foundation for Cancer Research, the Breast Cancer Research Foundation, and the Howard Hughes Medical Institute. Development and testing of the ^LPCTC-iChip was also supported by a generous gift from Ken and Grace Solinsky. AM was also supported by K25HL169816 from the National Heart, Lung, and Blood Institute. R.B. was supported by the Charles A. King Trust/Sara Elizabeth O'Brien Trust.

## Author contributions

A.M., D.A.H., S.M., and M.T. developed the concept. A.M., D.T.M., D.A.H., S.M., S.H., and M.T. designed experiments. D.T.M., D.A.H., S.M., and M.T. supervised the project. A.M., S.H., T.D., R.B., J.F.E., B.S.W., Q.E.C., V.R.P., A.D., E.A., K.A.G., K.O., J.E.K., S.Y.G., J.K., J.W., and M.S.L. conducted experiments and analyzed data. Y.M. and L.T.N. provided reagents. I.G., P.C., R.J.S., A.B., D.S.M., L.V.S., R.J.L., J.W.F., D.T.T., and P.A.R.B. provided clinical research samples and gave technical support. A.M. and S.H. contributed equally to this work.

## Competing interests

Massachusetts General Hospital has been granted patent protection for the inertial separation array (US20210370298A1, US11898209B2, and EP3560591B1) and inertial focusing technologies (US20190160465A1). Massachusetts General Hospital has a patent under review based on this work (US20230033651A1). MT, DAH, SM, and DTT are co-founders of TellBio, a biotechnology company commercializing the CTC-iChip technology using small blood volumes, which is distinct from the high throughput ^LPCTC-Chip technology and was not used in this work. All authors' interests were reviewed and managed by Massachusetts General Hospital and Mass General Brigham in accordance with their competing interests policies.
