## [Peer Review file · Nature Communications]

Tumor cell-based liquid biopsy using high-throughput microfluidic enrichment of entire leukapheresis product

Corresponding Author: Dr Mehmet Toner

Version 0:

Reviewer comments:

Reviewer #1

(Remarks to the Author)

In this manuscript by Mishra and colleagues, the authors described the application of their recently developed microfluidic technology to isolate and characterize circulating tumor cells (CTCs) from patient-derived leukapheresis products. The technology, which has been recently published, utilizes high-flow channels and amplification of cell sorting forces through magnetic lenses to achieve CTC isolation from leukopak that contains concentrated leukocytes and CTCs from 1-5L of blood. The authors optimized the technology and applied it to 6 metastatic cancer patients across different cancer types to capture very high numbers of CTCs. The large quantity of CTCs then allowed robust downstream analyses including immunofluorescent staining, single cell CNV analysis, and matched single cell RNA-sequencing from the same cell with CNV analysis, which led to novel insights into the heterogeneity of CTCs not only across cancer types but also within a particular patient. This approach offers a solution to the limited clinical utility of CTCs due to its rarity at a given time in small blood draws. Therefore, the manuscript is of significant technical and translational advantages to the field. Since the technology underlying the major basis of the approach has been previously reported, the novelty will be strengthened by additional analyses of the large quantity of CTCs with regards to their biology or clinical applications.

Major comments:

1. There are examples of CTC clusters in Figure 4A. It will be helpful to the field to report the proportions of CTC clusters isolated from leukopak, given the increasing awareness of the importance of the CTC clusters.
2. The confirmation of the tumor origin through CNV analysis for many of the DN cells is quite intriguing. It will be helpful to further characterize the confirmed DN tumor cells, such as their size, morphology, and other features.
 - Are they larger in cell or nuclear size than other DN WBCs?
 - Do they express cytokeratin? Is the DN phenotype due to the low level of expression of EpCAM and PSMA but they would be picked up by cytokeratin staining? Do they show EMT features?
 - Are there potential markers associated with those DN cells that can be potentially used to help identify those cells in the future?
 - Are these specific for GU-1 or also found in other patients?
3. In the example patient, GU-1, CTCs showed 2 distinct clusters by transcriptional profiling. It will be interesting to see the correlation of such finding to potential clinical implications, or at least discussion about the potential usage of such finding.

Minor comments:

Supplemental Table S3 missed some probe info

Certain supplemental figures were mentioned out of order, eg Sup Fig. S2 and S3.

Reviewer #2

(Remarks to the Author)

The study by A. Mishra et al. introduces a two-module microfluidic device for filtering of cells from blood, in a two-step process using inertial microfluidics and magnetic field-aided cell separation. Large volumes of 100 mL of pre-processed blood (by leukapheresis) are processed within these devices, which is in stark contrast to the usually processed volumes on microdevices of only a few milliliters. This opens up the possibility to capture efficiently rare cells such as circulating tumor cells during blood filtering of patients. The study builds on previous devices from the group with significant advancements for the special needs at such high throughput processing and high cell concentration. The authors demonstrate the applicability

by showing the results (phenotyping and genome sequencing of the CTCs) from several patients, suffering from various cancer types. It is exciting to read the study and therefore, this reviewer recommend to publish this work.

The study contains several interesting side aspects, particularly in the design of the device. However, this is very briefly described, illustrated with very few images and hard to understand. While many design aspects are in previous publications, it would be still helpful if the reader could understand the device without reading all other publications. In particular, figure 3 and several supplementary figures should be revised, or divided into two figures or described separate in the SI, together with a more detailed description of the chip functioning and operation. For example, it is not clear,

- A)How the "interconnect" is connected
- B)What these additional side channels at the waste outlet are good for.
- C)What design is and what function the parallel channels have after the inertial separation array
- D)Where the 16 separation devices are located (Figure 3 a-B is cited, but it is not visible in these images)
- E)What are the dimensions of the device, channels, etc.
- F)Where the part depicted in Figure S3 is located in the all-over chip?
- G)How the second chip operates. Subfigure B is too small, many parts of the chip are not explained.
- H)What are "high permeability channels", mentioned in the text and Figure 3F?
- I)Where the channels shown in Figure S5 are located. This figure is mentioned only in the discussion. The operation remains unclear.

Some minor points:

- 1.Abstract: The final volume processed in the device is 100 mL (?). It reads in the abstract as if 5.83 liters are interrogated on the devices.
- 2.What is known about the cell survival at these high flow rates?
- 3.What happens with cell clusters, e.g. CTC adhering to WBC?
- 4.Is it a feasible and useful approach with respect to the costs and handling?

Reviewer #3

(Remarks to the Author)

The manuscript presents the clinical implementation of the LPCTC-iChip, a device initially described for its potential to detect and isolate CTCs from entire leukapheresis products. This highly innovative approach integrates microfluidic debulking of red blood cells and platelets with magnetic sorting, leveraging enhanced magnetic forces. Consequently, it enables the processing of large blood volumes, significantly surpassing existing technologies in the number of isolated CTCs. This advancement underscores the potential to clinically exploit leukapheresis for liter-scale blood analysis, providing clinically relevant insights through the examination of molecular targets on CTCs—capabilities that other circulating biomarkers, such as ctDNA, more commonly used in clinical settings, cannot offer. Due to their shared density characteristics with mononuclear cells, tumor cells can be efficiently collected during leukapheresis along with targeted mononuclear cells, allowing the analysis of liters of blood without harming patients. The feasibility of using leukapheresis for CTC collection from extensive blood volumes and its integration into clinical routines was initially demonstrated by Fischer et al., with subsequent validation by several independent studies. However, methods for efficient detection and isolation of CTCs from entire leukapheresis products (leukopaks) were lacking. The authors present with their new technology a solution for this challenging problem. The clinical implications of this work are profound, considering the potential to harvest CTCs from the blood of nearly every advanced cancer patient, according to prior calculations from Coumans and colleagues (PMID: 23014524) and the research group involved in this study. Notably, leukapheresis offers a less invasive alternative compared to traditional biopsies or surgical procedures used for diagnostic purposes and can be more easily repeated for longitudinal monitoring. Another significant strength of this study lies in the comprehensive subsequent analyses of CTCs it enables, including profiling of protein expression (imaging), mRNA, and extensive genome characteristics at the single-cell level. Such comprehensive molecular and imaging data reinforce the utility of CTCs as a versatile surrogate for tissue-based clinical diagnostics. On the other hand, these data (particularly for the prostate cancer case) provide novel insights into CTC biology. The engineering and implementation of the LPCTC-iChip by this team could be a transformative advancement for liquid biopsy technology, facilitating the integration of CTC analysis into clinical practice—a progress previously hindered by the low detection rates typical of conventional 10 mL blood samples. This technology has the potential to revolutionize how liquid biopsies are conducted, making it a critical tool for non-invasive cancer diagnostics and therapy monitoring.

The manuscript is well-written and structured, methodologically sound including appropriate controls, and addresses a critical gap in cancer diagnostics and treatment monitoring. I believe that this manuscript, which describes scientifically significant work with high clinical relevance, is well-suited for Nature Communications. However, before publishing, the authors should address/clarify some points:

* It should be more clearly illustrated that leukapheresis has been systematically tested—initiated by the work of Fischer and colleagues (PMID: 24065821)—exclusively for the purpose of CTC enrichment, and should mention/discuss approaches to process larger portions/entire products in view of their more advanced technology (e.g., PMID: 30006930, PMID: 36425924, PMID: 33932725, PMID: 38720314).

* Since leukapheresis is introduced and applied for therapeutic purposes, I suggest using the term "diagnostic

leukapheresis" (DLA), which was introduced by Fischer and colleagues and has since been used by other independent groups investigating leukapheresis for CTC detection (e.g. PMID: 36425924, PMID: 34848855).

* The discussion effectively ties the results to broader implications and future applications. However, consider expanding on how this technology could integrate into current clinical workflows and its cost-effectiveness compared to existing biopsy methods. Please provide a more detailed discussion on this.

* A recent study by Rieckmann and colleagues reports (conventional) depletion of entire DLA products combined with transcriptomic profiling in NSCLC (PMID: 38720314). The authors should discuss this work.

* Another recent study (PMID: 38438456) presented a more systematic application and analysis of diagnostic leukapheresis in 60 patients with pancreatic cancer. It was demonstrated that despite increased detection, CTCs leveraged prognostic data. This recent study (analyzing only 5-10% of the leukapheresis product) should be discussed by the authors since it enforces the notion of the potential benefit of their technology for earlier tumor stages. On the other hand, such systematic analysis in larger patient cohorts reveals a weakness of the current study: it remains unclear at what frequency the new approach fails to detect/analyze CTCs. Will there be CTC-negative patients in advanced cancer cases? The authors should discuss this in more detail.

* The uveal melanoma cases appear very interesting, and I wonder whether the authors could also present genomic data on the circulating melanoma cells.

* Have the authors measured the CTC counts in a 10 mL blood sample using their chip technology to compare this to the DLA product? What is the efficiency of the CTC detection from DLA samples compared to peripheral blood samples (to determine the potential CTC loss)?

* The potential limitations of the study/technology should be discussed more comprehensively to provide a more balanced view.

* Please provide the working concentrations of the antibodies listed in Table S2.

Version 1:

Reviewer comments:

Reviewer #1

(Remarks to the Author)

The authors have fully addressed my comments. The revised manuscript is stronger and even more exciting.

Reviewer #2

(Remarks to the Author)

Thanks for the responses and clarification. The revised figures and added information are helpful for understanding the method. I recommend the publication of the study.

Reviewer #3

(Remarks to the Author)

The authors have sufficiently addressed all the previously raised issues. I believe that this manuscript is now ready for publication in Nature Communications.

RESPONSE TO REVIEWERS' COMMENTS

Reviewer #1 General Comments

In this manuscript by Mishra and colleagues, the authors described the application of their recently developed microfluidic technology to isolate and characterize circulating tumor cells (CTCs) from patient-derived leukapheresis products. The technology, which has been recently published, utilizes high-flow channels and amplification of cell sorting forces through magnetic lenses to achieve CTC isolation from leukopak that contains concentrated leukocytes and CTCs from 1-5L of blood. The authors optimized the technology and applied it to 6 metastatic cancer patients across different cancer types to capture very high numbers of CTCs. The large quantity of CTCs then allowed robust downstream analyses including immunofluorescent staining, single cell CNV analysis, and matched single cell RNA-sequencing from the same cell with CNV analysis, which led to novel insights into the heterogeneity of CTCs not only across cancer types but also within a particular patient. This approach offers a solution to the limited clinical utility of CTCs due to its rarity at a given time in small blood draws. Therefore, the manuscript is of significant technical and translational advantages to the field. Since the technology underlying the major basis of the approach has been previously reported, the novelty will be strengthened by additional analyses of the large quantity of CTCs with regards to their biology or clinical applications.

Authors' response

We thank the reviewer for their time and insightful summary of our work. We welcome the opportunity to strengthen our manuscript by further exploring the biological and clinical implications of the large quantities of CTCs as outlined below:

Specific Comment 1

There are examples of CTC clusters in Figure 4A. It will be helpful to the field to report the proportions of CTC clusters isolated from leukopak, given the increasing awareness of the importance of the CTC clusters.

Authors' comment

We and others have previously shown that CTC clusters are a rare, but highly metastatic-competent subset of CTCs in patients with metastatic cancer. In general, we need differently designed microfluidic devices using size-based flow separation to fully capture CTC clusters¹, and we have previously published such devices. However, they are not as well suited for high throughput "leukapheresis" processing, and they miss the vast majority of CTCs that are single cells. Despite these considerations, we have, in fact, identified a significant number of CTC clusters coming through the device in GU-1 patient, ranging in size from 2-5 cancer cells. Detection of CTC clusters and their size distribution depends on the biology of the individual tumor, as well as the leukapheresis settings and, ultimately, the microfluidic device.

On page 7 of the revised manuscript, we have added the following sentence: "In GU-1, we also detected 3,256 clusters of CTCs, ranging from 2-5 cancer cells tethered together in circulation (**Fig. 4A**) and in TNBC-1 we detected 80 two-cell CTC clusters."

Specific Comment 2a and 2b

The confirmation of the tumor origin through CNV analysis for many of the DN cells is quite intriguing. It will be helpful to further characterize the confirmed DN tumor cells, such as their size, morphology, and other features. Are they larger in cell or nuclear size than other DN WBCs? Do they express cytokeratin? Is the DN phenotype due to the low level of expression of EpCAM and PSMA but they would be picked up by cytokeratin staining? Do they show EMT features?

Authors' comment

We thank the reviewer for this insightful and important question, to which we devoted considerable attention. With further analysis and with the addition of another patient with metastatic prostate cancer (GU-2), we have found that the “double negative” cells (DN-CTCs: EpCAM/CK negative and leukocyte marker negative) are fascinating. They harbor the same clonal CNV patterns as classical prostate CTCs but lack epithelial/prostate lineage markers and express neuroendocrine markers. Moreover, their sizes overlap with WBCs; they are significantly smaller compared to that of EpCAM/CK positive CTCs. We propose that these represent lineage transformation from epithelial prostate cancer to a small-cell neuroendocrine lineage. Specific GSEA signatures that are increased in small DN cells include: ion channels, neuroendocrine pathways, and mesenchymal markers (some cells do retain expression of cytokeratin KRT8 and KRT18).

Such transformations have been identified in rare prostate tumor biopsies and are thought to contribute to acquired drug resistance and prostate cancer progression. The true frequency of such lineage transformation is unknown. The idea that such small cell/neuroendocrine subpopulations “quietly coexist” within patients with common forms of prostate cancer (and no evidence of such transformation by biopsy or other routine clinical tests) is of considerable interest.

We now devote an entire section of the results to this observation (page 10: Enrichment of small epithelial-negative prostate CTCs with neuroendocrine features”, and we have added additional data to Figure 7 (RNA profiling of all CTCs, including DN cells). We have also added an entirely new Figure 8 dedicated to in-depth analysis of the DN cell expression patterns and cell size data.

Specific Comment 2c and 2d

Are there potential markers associated with those DN cells that can be potentially used to help identify those cells in the future? Are these specific for GU-1 or also found in other patients?

Authors' comment

As suggested by the author, DN-CTCs would be missed by any CTC enrichment technology that relies on either size-based separation or EpCAM-capture, yet they may present a major clinically relevant population marking tumor evolution in metastatic prostate cancer. The use of specific neuroendocrine markers (Synaptophysin (SPY), EZH2, and Chromogranin A (CHGA)) to stain these cells would identify their presence.

We have found DN-CTCs in both GU-1 and GU-2 patients. Of note, GU-1 had a metastatic tumor biopsy with features of neuroendocrine differentiation (NED) consistent with the presence of CTCs with

neuroendocrine markers. However, the biopsy of GU-2 did not show any clinical evidence of NED but harbored DN CTCs, which express markers of NED. Together, we believe that NED may be more common in prostate cancer than currently assumed. The reviewer is correct that this kind of analysis may reveal previously unappreciated lineage transformations in other types of cancers as well.

Specific Comment 3

In the example patient, GU-1, CTCs showed 2 distinct clusters by transcriptional profiling. It will be interesting to see the correlation of such finding to potential clinical implications, or at least discussion about the potential usage of such finding.

Authors' comment

We agree with the reviewer that RNA-based identification of CTC subpopulations is of significant clinical interest, all the more so now that we have added patient GU-2, which also has distinct subpopulations (albeit different from those in GU-1). In the revised manuscript, we show that of the two CTC populations identified in GU-1, Cluster-1 is enriched in FGFR and AR signaling pathways, while Cluster-2 is enriched for JAK/STAT inflammatory signaling pathways (revised Fig. 7A-C). In GU-2, Cluster-1 is enriched for pathways associated with metastatic Castrate Resistant Prostate Cancer (mCRPC) progression, including AR, MYC, and oxidative phosphorylation, whereas Cluster 2 shows elevated gene signatures associated with ion channels and neuroendocrine differentiation (revised Fig. 7D-F). Together, these findings illustrate extensive intra- and inter-patient diversity in prostate tumor cell populations in castration-resistant disease. Interestingly, both GU-1 and GU-2 harbored small DN CTCs that express neuroendocrine markers. While the metastatic biopsy of GU-1 had features of NED, tissues from GU-2 did not suggest that NED might be prevalent. These two cases are, of course, too few to draw firm conclusions about broad clinical implications. We emphasize these findings in the revised manuscript (results and discussion sections) and stress the need to analyze more cases to better define the clinical significance of such findings.

Minor comments

Supplemental Table S3 missed some probe info

Certain supplemental figures were mentioned out of order, eg Sup Fig. S2 and S3.

Authors' comment

We apologize for these omissions. We have added the information for all the probes in Table S3 and in the Supplementary Information file (page 17).

We have also corrected the order of supplemental figures, as referenced in the text.

We thank the reviewer once again for their time and effort in reviewing our manuscript.

Reviewer #2 General Comments

The study by A. Mishra et al. introduces a two-module microfluidic device for filtering of cells from blood, in a two-step process using inertial microfluidics and magnetic field-aided cell separation. Large volumes of 100 mL of pre-processed blood (by leukapheresis) are processed within these devices, which is in stark contrast to the usually processed volumes on microdevices of only a few milliliters. This opens up the possibility to capture efficiently rare cells such as circulating tumor cells during blood filtering of patients. The study builds on previous devices from the group with significant advancements for the special needs at such high throughput processing and high cell concentration. The authors demonstrate the applicability by showing the results (phenotyping and genome sequencing of the CTCs) from several patients, suffering from various cancer types. It is exciting to read the study and therefore, this reviewer recommend to publish this work.

The study contains several interesting side aspects, particularly in the design of the device. However, this is very briefly described, illustrated with very few images and hard to understand. While many design aspects are in previous publications, it would be still helpful if the reader could understand the device without reading all other publications. In particular, figure 3 and several supplementary figures should be revised, or divided into two figures or described separate in the SI, together with a more detailed description of the chip functioning and operation.

Authors' response

We thank the reviewer for their time, positive comments, and a careful review of our manuscript. We are providing additional details and explanations to address the questions raised by the reviewer below:

Specific Comment 1

For example, it is not clear, how the “interconnect” is connected?

Authors' comment

Interconnect channels are located in a third layer (made of black plastic material in Fig. 3B), and these channels connect together waste outlet holes of 16 inertial separation arrays. The holes are connected by a via channel that exists in a third injection molded black plastic layer. This plastic layer was fused to the injection molded plastic chip, creating a single monolithic chip. Such third-layer-based interconnects allow outputs from multiple channels to be combined into a single outlet port. We now clarify these points in the text (page 5) and include Supplementary Figure S3, which specifically illustrates the concept of interconnect. For ease of review, this Figure is also reproduced below:

Legend Figure S3: *Interconnect channels allow outputs from multiple channels to be combined into a single outlet port. In the debulking chip, a third (black plastic) layer consists of interconnect channels, collecting waste outlets of 16 inertial separation arrays through 10 holes. This black plastic layer was fused to the top layer of the injection molded microfluidic plastic chip, creating a single monolithic chip. (A) Schematic diagram of the debulking chip with a layout of interconnecting channels. (B) An image of the debulking chip with a third layer made of black injection molded plastic. (C) A magnified image of an interconnecting channel (shown in green) collecting output from the multiple holes (shown in red).*

Specific Comments 2-4

What these additional side channels at the waste outlet are good for? What design is and what function the parallel channels have after the inertial separation array? Where the 16 separation devices are located (Figure 3 a-B is cited, but it is not visible in these images)?

Authors' comment

We thank the reviewer for raising these two important questions. Since it is very expensive to make molds for monolithic plastic chips, our disk design also included a few auxiliary devices on the chip. The additional side channels at the waste outlet are not directly involved in any function, and waste fluid does not flow through them. To avoid further confusion, we have deleted these channels from our schematic diagram in Fig. 3A, as shown below.

Similarly, parallel channels after the inertial separation array are additional devices that, in the context of this study, merely act as resistance channels. However, since the inertial separation array product flows through these channels, we decided to keep them on the schematic diagram.

As per the reviewer's instructions, we have also modified Figure 3 to enlarge the device in Fig. 3A and to clearly show the 16 inertial separation devices.

Specific Comment 5

What are the dimensions of the device, channels, etc?

Authors' comment

We have added a supplementary Figure S5 that details critical dimensions in debulking and MAGLENS chips. We show these below:

Legend Figure S5: *Critical dimensions of inertial separation array in debulking chip (A), sorting channels (B), and inertial cell concentrator in MAGLENS chip (C). All dimensions are in μm .*

Specific Comment 6

Where the part depicted in Figure S3 is located in the all-over chip?

Authors' comment

The Filter chip depicted in the updated Figure S2 (previously Figure S3) is placed directly inline before the high-throughput magnetic sorter for removing large clots or aggregates of cells. We have modified Figure S3, as shown below, to illustrate it more clearly. Please refer to Figure S2A on page 3 in the supplementary information file. This is represented below:

Figure S2: Filter chip. (A) This chip was used in-line with the magnetic sorter for removing large clots or aggregates of cells. (B) A schematic diagram of the filter chip. The inset shows streak images of fluorescently labeled WBCs flowing through the filter.

Specific Comment 7-8

How the second chip operates. Subfigure B is too small, many parts of the chip are not explained. What are “high permeability channels”, mentioned in the text and Figure 3F?

Authors' comment

As per the reviewer’s recommendation, we have increased the size of the second chip in Figure 3 to display various elements of the chip more clearly. We have also added extra details to explain various components of the magnetic sorter (MAGLENS Chip) on page 6 (reproduced below):

“The flow throughput of a continuous magnetic flow sorter is directly dependent on the magnetic field gradient in the deflection channels. Therefore, we incorporated high-permeability channels filled with soft magnetic iron particles. These channels act as “micromagnetic lenses” and amplify the magnetic field gradient 35-fold. This high magnetic gradient successfully enabled increased throughput in patient-derived leukopaks. Magnetic lenses create a field gradient as high as 15,400 T/m, compared to 440 T/m in the conventional CTC-iChip design^{2,3} (Sup Fig. S4), with each MAGLENS sorter processing a leukopak at a flow throughput of 48 mL/hour (3 billion cells/hour), 60-times higher than the conventional CTC-iChip.

The magnetic lenses are lithographically positioned 70 to 80 micrometers from the sorting channels (Sup Fig. S5).

Given the very high number of WBCs to be depleted from patient-derived leukopaks and their heterogeneous expression of leukocyte markers, we applied a cascaded two-stage magnetic sorting system. In stage I, cells tagged with >10 beads are deflected, and in stage II, WBCs labeled with 1 or greater magnetic beads are removed³ (Fig. 3F-H). Fig. 3I shows streak images from modeling studies where green-fluorescent WBCs are efficiently sorted from red-fluorescent CTCs. Effective depletion of WBCs carrying fewer magnetic beads requires stage II, where the flow rate is reduced through on-chip concentration to enable the depletion of cells labeled with a single bead. In stage I, the two asymmetric serpentine channels inertially focus cells in a single file by balancing shear-induced lift and Dean flow-based drag forces (Fig. 3F). The inertial focusing minimizes the possibility of WBCs colliding with a CTC and pushing it toward the center of the channel (discarded waste) during sorting (Fig. 3I). Furthermore, as leukapheresis product volumes are large, isolating CTCs within a smaller volume is essential. The inertial cell concentrator^A allows the final product to be concentrated 11-fold. Both the co-flow in stage I and the inertial concentrators in stage II ensure that cells remain close to the side walls, where the magnetic force is strongest due to proximity to the magnetic lenses (Sup Fig. S4). Overall, the total flow rate across stages I and II of the MAGLENS chip, including both leukopak sample and the added buffer, is 168 mL/hour.”

Specific Comment 9

What are “high permeability channels”, mentioned in the text and Figure 3F?

Authors’ comment

High (magnetic) permeability channels are filled with magnetically soft iron particles. These channels act as “micromagnetic lenses” and amplify the magnetic field gradient 35-fold. The micromagnetic lenses create a field gradient as high as 15,400 T/m compared to 440 T/m created by the conventional CTC-iChip arrangement^{2,3,5}. This is also described in the answer to the previous comment. We have modified Figure 3F to clearly illustrate that “high permeability channels” are “magnetic lenses.”

Specific Comment 10

Where the channels shown in Figure S5 are located. This Figure is mentioned only in the discussion. The operation remains unclear.

Authors’ comment

Inertial cell concentrators are located on the magnetic sorter chip in Figure 3F. This is now discussed on page 6 of the revised manuscript with the following text:

“Effective depletion of WBCs carrying fewer magnetic beads requires stage II, where the flow rate is reduced through on-chip concentration to enable the depletion of cells labeled with a single bead.”

“Furthermore, as leukapheresis product volumes are large, isolating CTCs within a smaller volume is essential. The inertial cell concentrator^A allows the final product to be concentrated 11-fold.”

The clogging aspect of these devices was discussed in the discussion section, but it is now also mentioned in the results section as well (Figure S6) and page 6. As below:

“To further optimize the processing of patient-derived leukopaks, we added DNase at 100 units/mL to the cell suspension to digest any neutrophil extracellular traps (NETs), which cause fouling of the microfluidic features (Sup Fig. S6).”

Minor Comments

1. *Abstract: The final volume processed in the device is 100 mL (?). It reads in the abstract as if 5.83 liters are interrogated on the devices.*

Authors' comment

100 mL leukapheresis products are collected from a mean human blood volume of 5.83 liters. The “interrogation” relates to the amount of patient blood volume from which CTCs are enriched versus the “processing,” which is the volume of concentrated leukopak flow through the Chip.

We have rephrased the sentence:

“Here, we apply this technology to analyze patient-derived leukapheresis products, interrogating a mean blood volume of 5.83 liters from seven patients with metastatic cancer.”

2. *What is known about the cell survival at these high flow rates?*

Authors' comment

In our previous PNAS manuscript³, we spiked cells from cultured CTCs into healthy donor leukopaks, processed these through our microfluidic devices, and tested the enriched CTCs for *in vitro* culture to assess whether the microfluidic devices damage the proliferative properties of the isolated CTCs. Enriched CTCs from the product proliferated comparably with control cells that were kept in a buffer solution (Fig. 4D in Mishra et al.³). The current system in this manuscript operates at similar shear stress levels, and we note that in previous publications, we have also generated patient-derived cultured CTCs using similar shear stress levels⁶, as well as mouse xenograft models⁷. Nonetheless, we acknowledge that a subset of damaged or pre-apoptotic CTCs may not survive the processing.

3. *What happens with cell clusters, e.g. CTC adhering to WBC?*

Authors' comment

We have shown that patient GU-1 has a large number of CTC clusters (ranging in size from 2 to 5 cells tethered together in the blood). Fig. 4A shows clusters of different morphologies isolated from this patient. We now mention this in the revised text in the results section on page 7. Heterotypic clusters of CTCs and WBCs will be lost during isolation due to the removal of WBCs with immunomagnetic beads conjugated with antibodies targeting CD45, CD66B, and CD16. Other devices, built by our lab and others, would be needed to identify such heterotypic clusters (based on positive selection for CTC markers rather than depletion of WBC markers)⁸.

4. *Is it a feasible and useful approach with respect to the costs and handling?*

Authors' comment

While diagnostic leukapheresis, coupled with high-throughput CTC enrichment, presents a strategy for clinical deployment of a whole-cell-based liquid biopsy, apheresis requires specialized clinical facilities, which imposes a practical hurdle. We have clarified this in the discussion section on page 14, stating that leukapheresis-based CTC analysis is “minimally invasive” as opposed to “noninvasive,” and it will be ideally suited for specific clinical decision points rather than general screening approaches. In the future, we aim to study whether sampled blood volume may be reduced further, particularly in cancer types with high CTC burdens. With respect to the engineering devices, the overall cost of Chips, reagents, and sequencing can be reduced by scale-up. We, therefore, feel that once the principle is established, individual clinical applications are readily implementable.

We thank the reviewer once again for their time and effort in reviewing our manuscript.

Reviewer #3 General Comments:

The manuscript presents the clinical implementation of the LPCTC-iChip, a device initially described for its potential to detect and isolate CTCs from entire leukapheresis products. This highly innovative approach integrates microfluidic debulking of red blood cells and platelets with magnetic sorting, leveraging enhanced magnetic forces. Consequently, it enables the processing of large blood volumes, significantly surpassing existing technologies in the number of isolated CTCs. This advancement underscores the potential to clinically exploit leukapheresis for liter-scale blood analysis, providing clinically relevant insights through the examination of molecular targets on CTCs— capabilities that other circulating biomarkers, such as ctDNA, more commonly used in clinical settings, cannot offer.

Due to their shared density characteristics with mononuclear cells, tumor cells can be efficiently collected during leukapheresis along with targeted mononuclear cells, allowing the analysis of liters of blood without harming patients. The feasibility of using leukapheresis for CTC collection from extensive blood volumes and its integration into clinical routines was initially demonstrated by Fischer et al., with subsequent validation by several independent studies. However, methods for efficient detection and isolation of CTCs from entire leukapheresis products (leukopaks) were lacking. The authors present with their new technology a solution for this challenging problem. The clinical implications of this work are profound, considering the potential to harvest CTCs from the blood of nearly every advanced cancer patient, according to prior calculations from Coumans and colleagues (PMID: 23014524) and the research group involved in this study.

Notably, leukapheresis offers a less invasive alternative compared to traditional biopsies or surgical procedures used for diagnostic purposes and can be more easily repeated for longitudinal monitoring. Another significant strength of this study lies in the comprehensive subsequent analyses of CTCs it enables, including profiling of protein expression (imaging), mRNA, and extensive genome characteristics at the single-cell level. Such comprehensive molecular and imaging data reinforce the utility of CTCs as a versatile surrogate for tissue-based clinical diagnostics. On the other hand, these data (particularly for the prostate cancer case) provide novel insights into CTC biology.

The engineering and implementation of the LPCTC-iChip by this team could be a transformative advancement for liquid biopsy technology, facilitating the integration of CTC analysis into clinical practice—a progress previously hindered by the low detection rates typical of conventional 10 mL blood samples. This technology has the potential to revolutionize how liquid biopsies are conducted, making it a critical tool for noninvasive cancer diagnostics and therapy monitoring.

The manuscript is well-written and structured, methodologically sound including appropriate controls, and addresses a critical gap in cancer diagnostics and treatment monitoring. I believe that this manuscript, which describes scientifically significant work with high clinical relevance, is well-suited for Nature Communications. However, before publishing, the authors should address/clarify some points.

Authors' comment

We thank the reviewer for these positive comments. We agree that diagnostic leukapheresis coupled with comprehensive single-cell analysis has the potential to not only replace biopsy but even surpass it in certain aspects.

Specific Comment 1

It should be more clearly illustrated that leukapheresis has been systematically tested—initiated by the work of Fischer and colleagues (PMID: 24065821)—exclusively for the purpose of CTC enrichment and should mention/discuss approaches to process larger portions/entire products in view of their more advanced technology (e.g., PMID: 30006930, PMID: 36425924, PMID: 33932725, PMID: 38720314).

Authors' comment

We apologize for the omission of these references. We have added these to the revised Discussion (page 11) with a more detailed review of previous work focused on both the original concept of leukapheresis for CTC analyses and various analytical approaches to leukapheresis-derived CTCs, including all the references listed above and additional references that are pertinent.

Specific Comment 2

The discussion effectively ties the results to broader implications and future applications. However, consider expanding on how this technology could integrate into current clinical workflows and its cost-effectiveness compared to existing biopsy methods. Please provide a more detailed discussion on this.

Authors' comment

We agree with the reviewer that leukapheresis-based CTC analyses may have clinical applications and, in some cases, supplant biopsy (especially in prostate cancer patients with bone metastasis) or other invasive procedures for diagnostic purposes. In the revised manuscript, we have referenced previous literature to illustrate this. We also provide more information (with the addition of patient GU-2) as to the type and quality of molecular insights that could be derived from CTCs by including the comparison of mutational profiles of metastatic biopsies, ctDNA and CTCs from the GU-1 and GU-2 (Figure 6). However, we would like to note that additional studies with a larger cohort of patients would be needed to draw strong conclusions about clinical workflows.

In terms of costs, the prototype devices described here are, of course, expensive and not yet suitable for broad dissemination and applications. However, once clinical utility is demonstrated and high throughput manufacturing is enabled, the cost of goods will precipitously decline. We feel it is too early to predict the cost of this approach as a clinical test, but we are very much aligned with the reviewer as to their ultimate importance.

Specific Comment 3

A recent study by Rieckmann and colleagues reports (conventional) depletion of entire DLA products combined with transcriptomic profiling in NSCLC (PMID: 38720314). The authors should discuss this work.

Authors' comment

We apologize for missing this reference and have now added it to the discussion. We note the importance of simultaneously typing single cells for both DNA (CNV) and RNA. It unequivocally shows the transcriptional profiles derived from bona fide CTCs. This is particularly important given the

significant variability we observe in purified marker-positive and -negative CTC populations. It is our hope that similar technologies will be transformative in enabling high throughput molecular characterizations.

Specific Comment 4

Another recent study (PMID: 37957606) presented a more systematic application and analysis of diagnostic leukapheresis in 60 patients with pancreatic cancer. It was demonstrated that despite increased detection, CTCs leveraged prognostic data. This recent study (analyzing only 5-10% of the leukapheresis product) should be discussed by the authors since it enforces the notion of the potential benefit of their technology for earlier tumor stages. On the other hand, such systematic analysis in larger patient cohorts reveals a weakness of the current study: it remains unclear at what frequency the new approach fails to detect/analyze CTCs. Will there be CTC-negative patients in advanced cancer cases? The authors should discuss this in more detail.

Authors' comment

We have limited our own study to patients with metastatic cancer but share the reviewer's interest in extending such analyses to localized cancers as well. Pancreatic cancer is particularly complex in that even localized cancer is often unresectable or may have already been disseminated microscopically. The ultimate goal, of course, would be to diagnose early cancers at a time when they are curable. We have added the reference above. Based on our own data, we can only comment in general terms about the eventual application of our technology to early cancer detection. Similarly, more studies by us and others will be required to determine how many "false negative" results are observed in the setting of localized curable cancers.

Specific Comment 5

The uveal melanoma cases appear very interesting, and I wonder whether the authors could also present genomic data on the circulating melanoma cells.

Authors' comment

Uveal Melanoma patient cases are indeed interesting. This cancer type is rare (our institution has a large referral population), but they are of considerable clinical interest as the primary tumors metastasize to the liver, raising the question of microscopic metastasis at presentation. For both cases studied here, we showed melanoma-specific RNA expression data from the enriched product (CTCs and residual blood cells) using ddPCR. Unfortunately, these patients were among the first few recruited, and at that time, we were hampered by COVID-related logistical challenges and were not able to apply paired single-cell CNV and RNA-seq to these samples.

Specific Comment 6

Have the authors measured the CTC counts in a 10 mL blood sample using their chip technology to compare this to the DLA product? What is the efficiency of the CTC detection from DLA samples compared to peripheral blood samples (to determine the potential CTC loss)?

Authors' comment

Unfortunately, our protocol did not include the collection of a concurrent 10mL blood sample alongside the diagnostic leukapheresis sample. However, multiple papers in the literature have directly compared the CellSearch platform for either 7.5mL of blood versus leukapheresis samples, showing the expected large increase in collected CTCs. These include Fehm et al⁹., Lambros et al¹⁰., and Stoecklein et al¹¹., which are referenced in the text.

Specific Comment 7

The potential limitations of the study/technology should be discussed more comprehensively to provide a more balanced view.

Authors' comment

We now specifically acknowledge in the discussion (page 14) the limitations given the small number of patients studied here, the limited number of analyses, and the fact that this is an initial proof of concept study.

Specific Comment 8

Please provide the working concentrations of the antibodies listed in Table S2.

Authors' comment

We have added the working concentrations of all the antibodies listed in Table S2.

We thank all three reviewers for their valuable time, positive and insightful comments and hope that we have adequately addressed them.

References:

1. Edd, J. F. *et al.* Microfluidic concentration and separation of circulating tumor cell clusters from large blood volumes. *Lab Chip* **20**, 558–567 (2020).
2. Fachin, F. *et al.* Monolithic Chip for High-throughput Blood Cell Depletion to Sort Rare Circulating Tumor Cells. *Sci. Rep.* **7**, 10936 (2017).
3. Mishra, A. *et al.* Ultra-high throughput magnetic sorting of large blood volumes for epitope-agnostic isolation of circulating tumor cells. *Proc. Natl. Acad. Sci.* **117**, 1–35 (2020).
4. Martel, J. M. *et al.* Continuous Flow Microfluidic Bioparticle Concentrator. *Sci. Rep.* **5**, 11300

- (2015).
5. Karabacak, N. M. *et al.* Microfluidic, marker-free isolation of circulating tumor cells from blood samples. *Nat. Protoc.* **9**, 694–710 (2014).
 6. Yu, M. *et al.* Ex vivo culture of circulating breast tumor cells for individualized testing of drug susceptibility. *Science (80-.)*. **345**, 216–220 (2014).
 7. Drapkin, B. J. *et al.* Genomic and Functional Fidelity of Small Cell Lung Cancer Patient-Derived Xenografts. *Cancer Discov.* **8**, 600–615 (2018).
 8. Ozkumur, E. *et al.* Inertial focusing for tumor antigen-dependent and -independent sorting of rare circulating tumor cells. *Sci. Transl. Med.* **5**, 179ra47 (2013).
 9. Fehm, T. N. *et al.* Diagnostic leukapheresis for CTC analysis in breast cancer patients: CTC frequency, clinical experiences and recommendations for standardized reporting. *Cytom. Part A* **93**, 1213–1219 (2018).
 10. Lambros, M. B. *et al.* Single-Cell Analyses of Prostate Cancer Liquid Biopsies Acquired by Apheresis. *Clin. Cancer Res.* **24**, 5635–5644 (2018).
 11. Stoecklein, N. H. *et al.* Ultra-sensitive CTC-based liquid biopsy for pancreatic cancer enabled by large blood volume analysis. *Mol. Cancer* **22**, (2023).